# CTP and *parS* coordinate ParB partition complex dynamics and ParA-ATPase activation for ParABS-mediated DNA partitioning

James A Taylor[1†], Yeonee Seol[2], Jagat Budhathoki[1], Keir C Neuman[2], Kiyoshi Mizuuchi[1]*

[1]Laboratory of Molecular Biology, National Institute of Diabetes and Digestive and Kidney Diseases, National Institutes of Health, Bethesda, United States; [2]Biochemistry and Biophysics Center, National Heart, Lung, and Blood Institute, National Institutes of Health, Bethesda, United States

**\*For correspondence:**
kiyoshimi@niddk.nih.gov

**Present address:** †Alzheimer's Research UK Oxford Drug Discovery Institute, University of Oxford, Oxford, United States

**Competing interests:** The authors declare that no competing interests exist.

**Abstract** ParABS partition systems, comprising the centromere-like DNA sequence *parS*, the *parS*-binding ParB-CTPase, and the nucleoid-binding ParA-ATPase, ensure faithful segregation of bacterial chromosomes and low-copy-number plasmids. F-plasmid partition complexes containing $ParB_F$ and $parS_F$ move by generating and following a local concentration gradient of nucleoid-bound $ParA_F$. However, the process through which $ParB_F$ activates $ParA_F$-ATPase has not been defined. We studied CTP- and $parS_F$-modulated $ParA_F$–$ParB_F$ complex assembly, in which DNA-bound $ParA_F$-ATP dimers are activated for ATP hydrolysis by interacting with two $ParB_F$ N-terminal domains. CTP or $parS_F$ enhances the ATPase rate without significantly accelerating $ParA_F$–$ParB_F$ complex assembly. Together, $parS_F$ and CTP accelerate $ParA_F$–$ParB_F$ assembly without further significant increase in ATPase rate. Magnetic-tweezers experiments showed that CTP promotes multiple $ParB_F$ loading onto $parS_F$-containing DNA, generating condensed partition complex-like assemblies. We propose that $ParB_F$ in the partition complex adopts a conformation that enhances $ParB_F$–$ParB_F$ and $ParA_F$–$ParB_F$ interactions promoting efficient partitioning.

## Introduction

Faithful segregation of replicated chromosomes is essential for efficient proliferation of cells. Accordingly, many bacteria are equipped with active chromosome and plasmid partition systems belonging to the ParABS family (*Baxter and Funnell, 2014*; *Lutkenhaus, 2012*; *Vecchiarelli et al., 2012*). Basic ParABS systems comprise two proteins, ParA and ParB, and a centromere-like, cis-acting DNA element called *parS*. The ParA proteins of this family are ATPases with a characteristic deviant Walker-A motif (*Motallebi-Veshareh et al., 1990*) and bind non-specific DNA (nsDNA) in an ATP-dependent manner by forming a DNA binding-competent dimer (*Davey and Funnell, 1994*; *Leonard et al., 2005*; *Vecchiarelli et al., 2010*). Accordingly, ParA proteins localize to the bacterial chromosome (the nucleoid) *in vivo* (*Ebersbach and Gerdes, 2004*; *Hatano et al., 2007*; *Lim et al., 2014*).

ParB is typically a dimeric sequence-specific DNA binding protein that binds tightly to the *parS* consensus sequences that mark the DNA cargo to be partitioned (*Bouet et al., 2000*; *Mori et al., 1989*; *Pillet et al., 2011*; *Taylor et al., 2015*). Most ParABS systems have multiple copies of a ParB dimer binding consensus sequence that collectively constitute a *parS* site. F-plasmid has a *parS* sequence cluster (*parS_F*, also called *sopC*) composed of 12 repeats of a 16 bp consensus sequence, each separated by 27 base-pair spacer sequences (*Helsberg and Eichenlaub, 1986*). The ParBs of

known chromosomal and plasmid Par systems such as P1 and F bind to *parS via* a helix-turn-helix motif (*Schumacher and Funnell, 2005*; *Schumacher et al., 2010*). These HTH–ParB proteins also associate with several kilobases of DNA surrounding a *parS* site *in vivo* in a proximity-dependent manner without obvious sequence specificity (*Breier and Grossman, 2007*; *Murray et al., 2006*; *Rodionov et al., 1999*; *Sanchez et al., 2015*). This activity, known as ParB spreading, is believed to be essential for proper function of these systems (*Breier and Grossman, 2007*; *Graham et al., 2014*) and results in the formation of a large nucleo-protein complex (the partition complex) around the *parS* site on the DNA to be partitioned. Mutations in the *B. subtilis parB* gene blocking spreading and causing partition deficiency have been identified within the Box II region (GXRR) of the N-terminal domain (*Breier and Grossman, 2007*; *Graham et al., 2014*), a highly conserved motif among HTH–ParB homologues (*Yamaichi and Niki, 2000*). Recently, several groups reported that HTH–ParB proteins have CTPase activity and the Box II residues play key roles in CTP binding and hydrolysis, suggesting that ParB spreading is driven by an active process dependent on energy derived from CTP hydrolysis (*Jalal et al., 2020*; *Osorio-Valeriano et al., 2019*; *Soh et al., 2019*).

ParB interacts with ParA *via* its N-terminal region (*Ravin et al., 2003*) and activates nsDNA-bound ParA dimer's ATPase, releasing it from DNA (*Ah-Seng et al., 2009*; *Davis et al., 1992*; *Scholefield et al., 2011*; *Watanabe et al., 1992*). In the absence of ParB stimulation, the ATP turnover of ParA is low, typically around one ATP per hour (*Ah-Seng et al., 2009*; *Davis et al., 1992*; *Fung et al., 2001*; *Scholefield et al., 2011*). Because of this slow basal ATPase rate, and since the majority of cellular ParB molecules is concentrated at the partition complexes due to ParB spreading, ATP hydrolysis by ParA and dissociation from the nucleoid is expected to occur principally in the vicinity of partition complexes. Biochemical studies of P1 ParA ATPase showed a significant time delay before activation of ParA for DNA binding after ATP binding, predicting a significant free bulk-diffusion period for ParA before reactivation for nsDNA binding (*Vecchiarelli et al., 2010*). This, along with *in vivo* imaging observations (*Hatano et al., 2007*; *Ringgaard et al., 2009*), led to a prediction that the nucleoid proximal to a partition complex would become depleted of ParA (the ParA depletion zone) and the proposal of a diffusion-ratchet model for plasmid segregation by the ParABS system (*Vecchiarelli et al., 2010*).

The diffusion-ratchet model is based on the premise that the nucleoid-bound ParA-ATPase activation by plasmid-bound ParB generates a local ParA depletion zone on the nucleoid, forming a nucleoid-bound ParA concentration gradient around the partition complex (*Hu et al., 2017*; *Vecchiarelli et al., 2010*). The interaction of plasmid-bound ParB with the ParA gradient on the nucleoid results in a cargo position-dependent free-energy difference (*Sugawara and Kaneko, 2011*). Binding of ParB to ParA reduces the system free-energy, therefore moving the cargo to a higher ParA concentration lowers the system free-energy. This cargo position-dependent free-energy difference translates to a directional motive force on the ParB bound cargo. The generation of sufficient cargo motive force to overcome thermal diffusion was demonstrated in cell-free reconstitution experiments showing that a bead coated with *parS$_F$*-containing DNA is driven across an nsDNA-coated flow cell surface in the presence of ParA$_F$, ParB$_F$, and ATP (*Vecchiarelli et al., 2014*). In some *in vivo* time-lapse imaging experiments, ParA has been observed to undergo pole-to-pole oscillations along the length of a nucleoid with a partition complex chasing the receding edge of a ParA distribution zone on the nucleoid, further supporting the diffusion-ratchet model (*Hatano et al., 2007*; *Ringgaard et al., 2009*). Variations of this model have also been proposed (*Le Gall et al., 2016*; *Lim et al., 2014*; *McLeod et al., 2017*).

Generating persistent directional motion by the diffusion-ratchet model requires a balance between the ParA–ParB association/dissociation dynamics prior to ATP hydrolysis and the steps and kinetic parameters that govern ATP hydrolysis. For example, if every ParA–ParB association resulted in instantaneous ATP hydrolysis and dissolution of the ParA–ParB bonds linking the partition complex (cargo) and the nucleoid, no cargo driving force would result. Conversely, if each bond persisted too long, the cargo would remain immobile on the nucleoid (*Hu et al., 2017*). However, the detailed biochemical reaction steps leading to ParB activation of DNA-bound ParA-ATPase and the subsequent release of ParA have not been determined.

In order to advance our quantitative understanding of the ParABS system mechanism, we investigated the assembly and disassembly kinetics of ParA$_F$–ParB$_F$ complexes that form prior to ATP hydrolysis using F-plasmid ParA$_F$B$_F$S$_F$ (also called SopA/B/C) as a model system. Employing a TIRF microscopy-based nsDNA-carpet assay (*Vecchiarelli et al., 2013*), we first examined the

stoichiometry of the nsDNA-bound $ParA_F$–$ParB_F$ complexes that accumulate in the absence of ATP hydrolysis. We investigated impacts of different $ParB_F$-cofactors ($parS_F$ and CTP) or $ParB_F$ mutations that hinder $ParB_F$ dimerization, $parS_F$–binding, CTPase activity, or $ParA_F$ ATPase activation. We then studied how the same set of cofactors or $ParB_F$ mutations influenced $ParA_F$-ATPase activation by $ParB_F$. Our results showed that $ParB_F$ formed complexes with nsDNA-bound $ParA_F$ in which both $ParB_F$-interacting faces of the $ParA_F$ dimers were occupied by the N-terminal $ParA_F$-activation domain of $ParB_F$. All such complexes observed in the presence of ATPγS exhibited similar, slow dissociation kinetics from nsDNA compared to $ParA_F$ dimers in the absence of $ParB_F$ so long as both $ParB_F$-interacting faces of the $ParA_F$ dimers were occupied by $ParB_F$ N-terminal domains. Binding of $ParB_F$ N-terminal domains at both $ParB_F$ interaction faces is also necessary for efficient ATPase activation of nsDNA-bound $ParA_F$ dimers. Strikingly, the $ParB_F$ cofactors, CTP and $parS_F$, acted synergistically to accelerate the assembly of pre-ATP hydrolysis $ParB_F$–$ParA_F$ complexes on nsDNA. In addition, a magnetic tweezers-based DNA condensation assay revealed that stable DNA condensation by $ParB_F$ required both CTP and $parS_F$. These observations suggested that CTP and $parS_F$ promote both $ParA_F$–$ParB_F$ and $ParB_F$–$ParB_F$ interactions. The compaction of $parS_F$-containing DNA by $ParB_F$ in the presence of CTP recapitulated the salient features of the condensed ParB spreading partition complex observed *in vivo*, suggesting that $ParB_F$ in the presence of $parS_F$ and CTP closely reflects the functional state of $ParB_F$ in the partition complexes *in vivo*. Interestingly, $ParB_F$ in these conditions accelerated ATP turnover by the nsDNA-bound $ParA_F$ no more than twofold compared to $ParB_F$ without $parS_F$ and CTP, to a modest ~80 $h^{-1}$. These findings have important implications for the diffusion-ratchet model of F-plasmid partition by the ParAB*S* system.

## Results

Here, we investigated the ParA–ParB interactions that lead to accelerated ATP hydrolysis by ParA and how they are impacted by $parS_F$ and CTP. First, we addressed the nature of nsDNA-bound $ParA_F$–$ParB_F$ complexes. In the presence of ATP, $ParA_F$ forms DNA binding-competent dimers and binds the nsDNA-carpet without $ParB_F$. Upon forming a complex with $ParB_F$, $ParA_F$ becomes activated for ATP hydrolysis and the complex rapidly disassembles (*Vecchiarelli et al., 2013*), impeding characterization of the complex. Therefore, we studied DNA-bound $ParA_F$–$ParB_F$ complexes that accumulate prior to ATP hydrolysis by using non-hydrolysable ATPγS. $ParA_F$ is not expected to form DNA binding competent dimers efficiently in the presence of ATPγS based on the study of the closely related $ParA_{P1}$ ATPase (*Vecchiarelli et al., 2010*). However, we found $ParB_F$ promotes conversion of $ParA_F$ to a DNA binding competent state in the presence of ATPγS (see below). Hence, in this study, we infused $ParA_F$-eGFP and $ParB_F$-Alexa647, or other fluorescent-labeled $ParB_F$ variants, preincubated at room temperature for 10 min with ATPγS into an nsDNA-carpeted flow cell and quantified the densities of the two DNA-associated proteins by imaging the flow cell surface with TIRF microscopy (*Figure 1*). $ParA_F$ alone at the concentration used here (1 μM, all protein concentrations are expressed as monomer concentrations) does not efficiently bind DNA in the presence of ATPγS as expected, and only low-level steady state density (less than ~200 monomers per μm$^2$) of $ParA_F$-eGFP was detected on the DNA-carpet (*Figure 2A,B*). The observed DNA dissociation rate constant of $ParA_F$-ATPγS (~6 $min^{-1}$) is similar to that of $ParA_F$-ATP estimated by FRAP or by washing the flow-cell with nsDNA-containing buffer (~4.5–6 $min^{-1}$, *Vecchiarelli et al., 2013*). $ParB_F$-Alexa647 (2 μM) did not bind the DNA-carpet to a significant level in the 300 mM K-glutamate buffer used in this experiment.

### The N-terminus of $ParB_F$ alone enhances DNA binding activity of $ParA_F$

We first examined if the N-terminal $ParA_F$-activation domain of $ParB_F$ ($ParB_F^{1-42}$) alone can induce the $ParA_F$ conformational changes necessary for DNA binding in the presence of ATPγS. The activation domain includes Arginine 36, which is critical for activation of $ParA_F$ ATPase (*Ah-Seng et al., 2009*; *Leonard et al., 2005*), but lacks the CTPase (AA63-155), $parS_F$–binding (AA160-272), and dimerization (AA276-323) domains. For these experiments, we used $ParB_F^{1-42}$ fused to the N-terminus of mCherry ($ParB_F^{1-42}$-mCherry). This protein is a monomer in solution as judged by its elution profile on a gel filtration column (*Figure 2—figure supplement 1*). $ParB_F^{1-42}$-mCherry (10 μM) and $ParA_F$-eGFP (1 μM) bound the DNA-carpet together in the presence of ATPγS to a density of ~5000 monomers/μm$^2$ maintaining ~1:1 stoichiometry (*Figure 2C*). When washed with a buffer containing ATPγS, $ParB_F^{1-}$

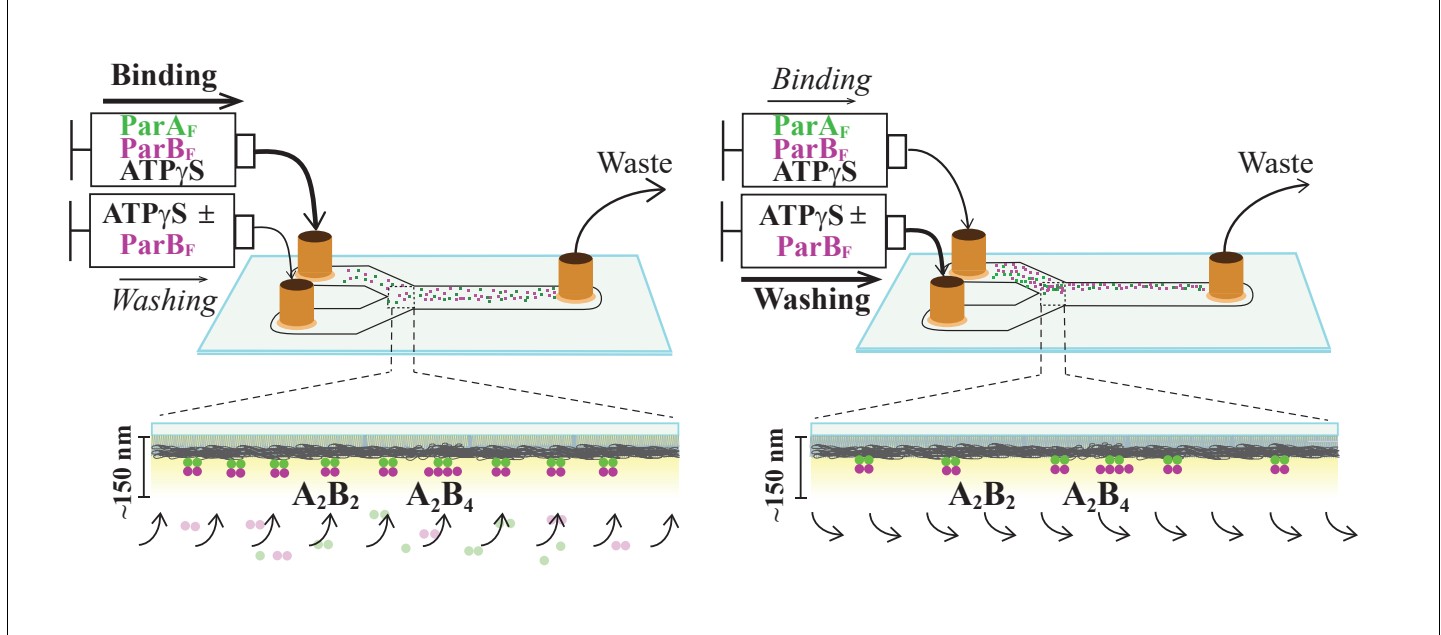

**Figure 1.** Schematic of flow cell setup for visualizing the binding and dissociation of fluorescent proteins on DNA-carpet. ParA$_F$-eGFP and ParB$_F$-Alexa647 proteins were flowed over a dense carpet of nsDNA attached to the supported lipid bilayer coated surface of a flow cell. TIRF microscopy permits selective detection of the DNA-carpet bound proteins. Sample solution and wash buffer, as specified for each experiment, were infused *via* two syringes at different infusion rates from separate inlets into a Y-shaped flow cell. A laminar boundary separates the two solutions downstream of the flow convergence point at the Y-junction. At the midpoint across the flow channel, downstream but close to the flow convergence point where the observations are made, the DNA-carpet area is exposed to the syringe content of the higher infusion rate. When the infusion rates of the two syringes are switched, the laminar boundary moves across the observation area and the solution flowing over the area switches. By switching the infusion rates of the two syringes repeatedly, multiple DNA-carpet-bound protein complex assembly and wash cycles can be recorded.

42-mCherry dissociated first, with an apparent dissociation rate constant of ~5.7 min$^{-1}$, closely followed (within a few seconds) by ParA$_F$-eGFP dissociation (*Figure 2D*). On the other hand, when the wash buffer contained 10 μM ParB$_F^{1-42}$-mCherry and ATPγS, both proteins dissociated together, significantly slower than ParA$_F$-eGFP bound to the DNA-carpet alone, with an apparent rate constant of ~0.9 min$^{-1}$, maintaining ~1:1 stoichiometry (*Figure 2E*). After essentially complete dissociation of ParA$_F$-eGFP from the DNA-carpet, 10 μM ParB$_F^{1-42}$-mCherry present in the wash solution showed no significant binding to the DNA-carpet, indicating low intrinsic affinity of this protein for DNA. FRAP measurements of ParA$_F$-eGFP—ParB$_F^{1-42}$-mCherry bound in steady state to the DNA-carpet in the presence of ATPγS also indicated rapid exchange of ParB$_F^{1-42}$-mCherry (*Figure 2—figure supplement 2*). Thus, at saturating ParB$_F^{1-42}$-mCherry concentration, a ParA$_F$—ATPγS dimer is bound by two molecules of ParB$_F^{1-42}$-mCherry occupying both sides of the ParA$_F$ dimer. The data also indicate that ParA$_F$ dimers adopt a state of slowed dissociation from nsDNA when both of the ParB$_F$-interacting faces are occupied by the ParB$_F$ N-terminal domain. The nsDNA dissociation rate constants of ParA$_F$—ParB$_F$ complexes (including those involving ParB$_F$ variants) and ParB$_F$:ParA$_F$ stoichiometry reported above and in the following sections are summarized in *Table 1*.

## ParA$_F$ ATPase activation requires binding of two copies of ParB$_F$ N-terminal domain to the ParA$_F$ dimer

ParB$_F^{1-42}$ stimulated ParA$_F$-ATPase (1 μM) with a clear sigmoidal ParB$_F^{1-42}$ concentration dependence and a half-maximum activation concentration of ~1.2 μM (*Figure 2F*; *Table 2*). Thus, monomeric ParB$_F^{1-42}$ appears to activate ParA$_F$-ATPase when it binds on both sides of the DNA-bound ParA$_F$ dimers. To test whether the observed sigmoidal concentration dependence is due to the monomeric nature of ParB$_F^{1-42}$, we prepared the N-terminal region of ParB$_F$ fused to mCherry and the nuclease activity deficient EcoRI$^{E111Q}$, ParB$_F^{1-42}$-mCherry-EcoRI$^{E111Q}$ (see *Figure 2—figure supplement 3A*). This construct, with expected EcoRI dimerization $K_D$ < 20 pM (*Modrich and Zabel, 1976*), efficiently

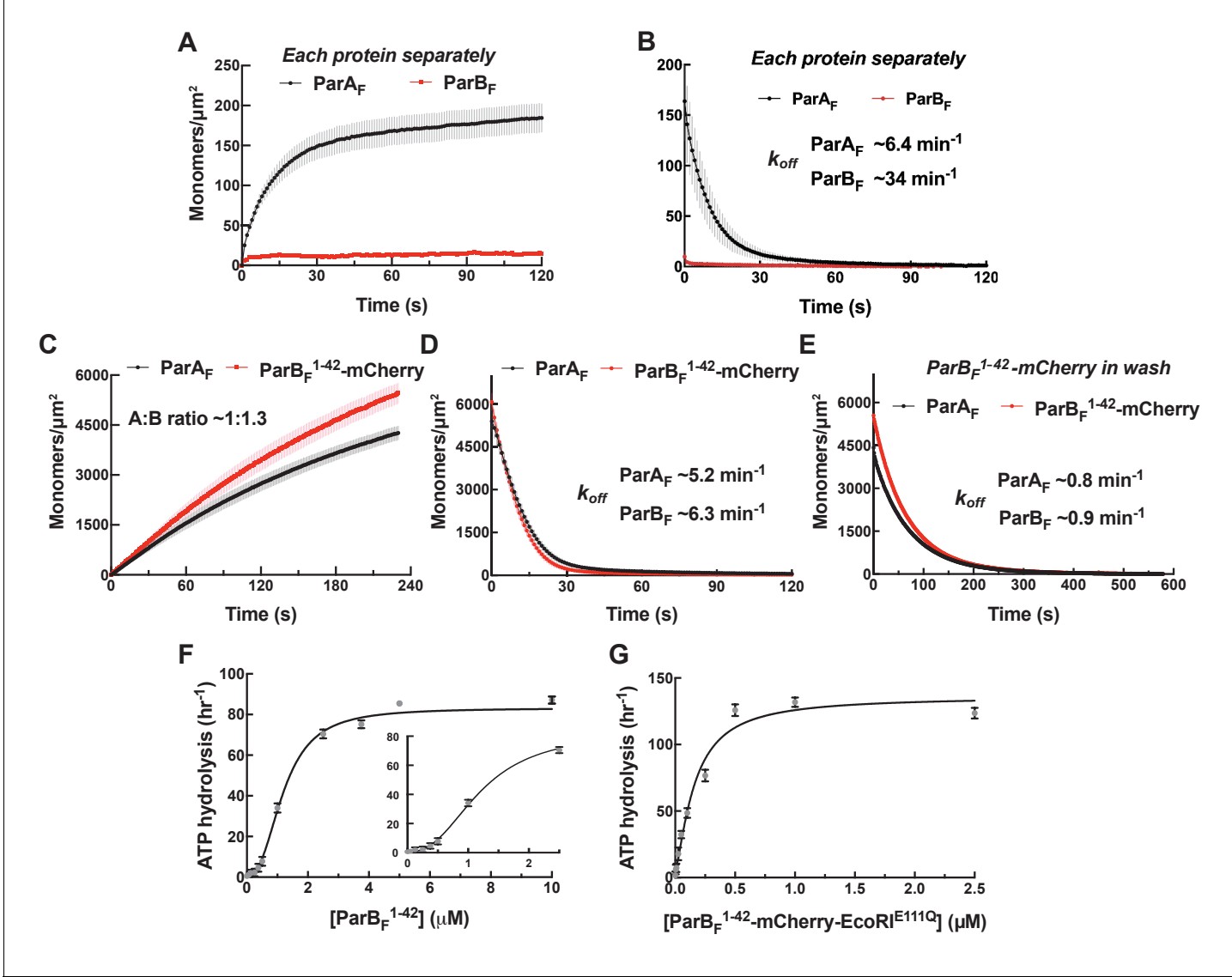

**Figure 2.** Monomeric ParB$_F^{1-42}$-mCherry can activate ParA$_F$ for nsDNA binding in the presence of ATPγS by forming a ~1:1 complex. Protein sample solution in the presence of ATPγS (1 mM) was infused into nsDNA-carpeted flow cell at a constant flow to monitor the protein binding to the nsDNA, and the sample solution was switched to a wash buffer containing ATPγS to monitor protein dissociation from nsDNA. (A, B) Binding to, and dissociation from, nsDNA of ParA$_F$-eGFP (1 μM) or ParB$_F$-Alexa647 (2 μM) were measured separately. (C) ParB$_F^{1-42}$-mCherry (10 μM) and ParA$_F$-eGFP (1 μM) preincubated with ATPγS were infused into the nsDNA-carpeted flow cell and (D) washed with buffer containing ATPγS. (E) The washing experiment of (D) was repeated with wash buffer containing ATPγS and ParB$_F^{1-42}$-mCherry (10 μM). For the parameters of the time courses of above experiments and subsequent experiments of the same type in this study, see *Table 1*. The ParB$_F$:ParA$_F$ ratio was calculated from carpet-bound densities of the two proteins measured in parallel, and summarized in *Table 1*. (F) ParA$_F$-ATPase activity (expressed as turnover rate per ParA$_F$ monomer) was measured in the presence of EcoRI-digested pBR322 DNA (60 μg/ml) and different concentrations of ParB$_F^{1-42}$. Inset shows a plot with expanded abscissa. (G) ParA$_F$-ATPase activity was measured as above in the presence of dimeric ParB$_F^{1-42}$-mCherry-EcoRI$^{E111Q}$. The parameters of ATPase stimulation curves in these and subsequent figures are summarized in *Table 2*.

The online version of this article includes the following figure supplement(s) for figure 2:

**Figure supplement 1.** Gel filtration column elution profile of ParB$_F^{1-42}$-mCherry.

**Figure supplement 2.** FRAP of ParA$_F$-eGFP and ParB$_F^{1-42}$-mCherry on DNA-carpet.

**Figure supplement 3.** Comparison of possible structural domain arrangements of artificially dimeric ParB$_F^{1-42}$-mCherry-EcoRI$^{E111Q}$.

**Figure supplement 4.** ParA$_F$ ATPase stimulation by ParB$_F^{1-42}$-mCherry-EcoRI$^{E111Q}$ is not influenced by the addition of DNA fragment containing EcoRI recognition sequence.

**Table 1.** Apparent disassembly or exchange rate constants (min$^{-1}$) and $ParB_F/ParA_F$ ratio from fits of TIRFM wash and FRAP experiments.

The apparent dissociation (or FRAP) rate constants ($k_{off}$) were obtained for individual time-trajectories by single-exponential curve fitting (except ** where the rate of the faster decay, (68 ± 0.5%) of a double-exponential fit is shown), and the mean and SEM for the set of independent experiments are shown (except * where standard deviation among non-independent repeats within an experiment is shown). (N is the number of separate experiments, with total number of binding/wash cycles for repeated data collection in parenthesis.) ParA:ParB ratios were calculated from the final phase of the individual association time-trajectories (except *** where it was based on the beginning part of the washing phase in the presence of $ParB_F^{1-42\ R36A}$), and the mean and SEM for the set of independent experiments are shown in italics. N.D., not done.

| Protein Measured | | $ParA_F$ | $ParA_F + ParB_F^{1-42}$ | | $ParA_F + ParB_F^{1-42\ R36A}$ | | FRAP | | $ParA_F + ParB_F$ | | $ParA_F + ParB_F + CDP$ | | $ParA_F + ParB_F + CTP$ | | $ParA_F + ParB_F^{R121A}$ | |
|---|---|---|---|---|---|---|---|---|---|---|---|---|---|---|---|---|
| | | $ParA_F$ | $ParA_F$ | $ParB_F^{1-42}$ | $ParA_F$ | $ParB_F^{1-42\ R36A}$ | $ParA_F$ | $ParB_F^{1-42}$ | $ParA_F$ | $ParB_F$ | $ParA_F$ | $ParB_F$ | $ParA_F$ | $ParB_F$ | $ParA_F$ | $ParB_F^{R121A}$ |
| - parS_F | $k_{off}$ | $6.4\pm 0.6$ N=4 (12) | $5.2\pm 0.2$ N=2 (9) | $6.3\pm 0.3$ N=2 (9) | $29\pm 2$ N=3 (9) | $52\pm 4$ N=3 (9) | $2.0\pm 0.08$ N=3 (11) | $15.3\pm 2.1$** N=3 (12) | $0.93\pm 0.07$ N=2 (6) | $0.89\pm 0.09$ N=2 (6) | N.D. | N.D. | $0.63\pm 0.04$ N=2 (7) | $0.65\pm 0.10$ N=2(7) | $0.89\pm 0.01$ N=3 (8) | $0.78\pm 0.05$ N=3 (8) |
| | A:B | | *1 : 1.27 ± 0.06 N=2 (8)* | | *1 : 0.92 ± 0.02*** N=3 (9)* | | | | *1 : 1.10 ± 0.13 N=3 (9)* | | | | *1 : 2.16 ± 0.15 N=3 (10)* | | *1 : 0.99 ± 0.01 N=3 (8)* | |
| + parS_F | $k_{off}$ | N.D. | N.D. | | N.D. | | N.D. | | $0.95\pm 0.04$ N=2 (6) | $0.82\pm 0.01$ N=2 (6) | $0.46\pm 0.02$ N=2 (7) | $0.53\pm 0.01$ N=2 (7) | $0.60\pm 0.08$ N=2 (5) | $0.83\pm 0.12$ N=2 (6) | $0.91\pm 0.01$ N=3 (8) | $0.78\pm 0.04$ N=3 (8) |
| | A:B | | N.D. | | N.D. | | | | *1 : 2.17 ± 0.40 N=3 (9)* | | *1 : 2.53 ± 0.07 N=2 (6)* | | *1 : 2.16 ± 0.27 N=2(6)* | | *1 : 1.12 ± 0.01 N=3 (12)* | |
| $ParB_F$ or $ParB_F^{1-42\ R36A}$ in wash | $k_{off}$ | N.D. | $0.84\pm 0.01$* N=1 (3) | $0.89\pm 0.02$* N=1 (3) | $1.8\pm 0.4$ N=3 (9) | $1.8\pm 0.4$ N=3 (9) | N.D. | | N.D. | | N.D. | | N.D. | | N.D. | |

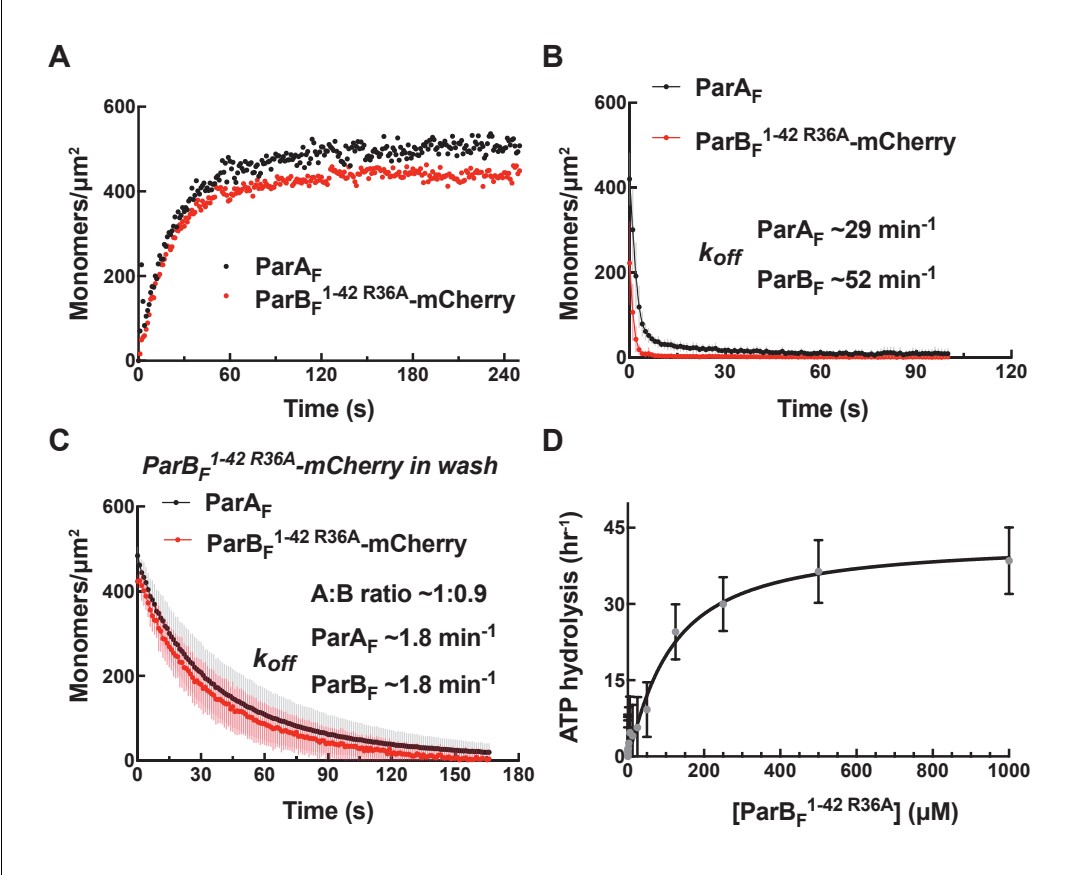

**Figure 3.** ParB$_F^{1-42\ R36A}$-mCherry dissociates faster from nsDNA-carpet-bound ParA$_F$-ATPγS dimer, and ParA$_F$-ATPase activation requires higher ParB$_F^{1-42\ R36A}$ concentration. (A) ParB$_F^{1-42\ R36A}$-mCherry (10 μM) and ParA$_F$-eGFP (1 μM) preincubated with ATPγS were infused into the nsDNA-carpeted flow cell and then (B) washed with buffer containing ATPγS. (C) The washing experiment of B was repeated with buffer containing ATPγS and ParB$_F^{1-42\ R36A}$-mCherry (10 μM). (D) ParA$_F$-ATPase activity was measured in the presence of EcoRI-digested pBR322 DNA (60 μg/ml) as a function of ParB$_F^{1-42\ R36A}$ concentration. See *Figure 2* legend and *Tables 1* and *2* for additional details.

activated ParA$_F$-ATPase at least to a similar maximum rate as ParB$_F^{1-42}$, but with a K$_{half}$ of ~0.15 μM, ~eight fold lower than ParB$_F^{1-42}$, and displayed no noticeable sigmoidal concentration dependence (*Figure 2G*). Potential binding of the inactive EcoRI domain to DNA did not appear to influence the ATPase activation properties of this construct; addition of EcoRI-binding DNA fragment in the reaction did not impact the ATPase activation (*Figure 2—figure supplement 4*). Based on these results, we conclude that both ParB$_F$-binding faces of a ParA$_F$ dimer must be occupied by ParB$_F$ N-termini for stimulation of its ATPase activity.

## ParB$_F^{1-42\ R36A}$ forms a rapidly disassembling complex with ParA$_F$ on the DNA carpet

An R36A mutation was reported to significantly compromise ParB$_F$'s ability to activate ParA$_F$'s ATPase (*Ah-Seng et al., 2009*). To test whether this mutation affected ParB$_F$'s ability to form a complex with ParA$_F$ we repeated the experiments shown in *Figure 2C–E* using ParB$_F^{1-42\ R36A}$-mCherry. ParB$_F^{1-42R36A}$-mCherry and ParA$_F$-eGFP bound with an approximately 1:1 stoichiometry, similar to ParB$_F^{1-42}$-mCherry but reached a steady-state density on the DNA-carpet of less than 10% of the density observed with ParB$_F^{1-42}$-mCherry (*Figure 3A*). When washed with buffer containing ATPγS, ParB$_F^{1-42\ R36A}$-mCherry dissociated first followed by ParA$_F$, similar to the results obtained with ParB$_F^{1-42}$-mCherry but ParB$_F^{1-42\ R36A}$-mCherry dissociated ~10-fold faster, followed by dissociation of ParA$_F$-eGFP within a few seconds (*Figure 3B*). When the wash buffer also contained 10 μM ParB$_F^{1-42\ R36A}$-mCherry the two proteins dissociated in parallel maintaining ~1:1 stoichiometry (*Figure 3C*). Together these observations indicate that ParB$_F^{R36A}$

**Table 2.** ATPase fit parameters.

ATPase measurements were performed with $ParA_F$ (1 µM) and different mutants of $ParB_F$, 60 µg/ml EcoRI-digested pBR322 DNA plus Scram- or $parS_F$-DNA fragment and CTP or CDP, as indicated in the column headings. Assays were repeated 'N' times, each data set of an assay was fit after subtraction of background measured without $ParA_F$ to a modified Hill equation: $v - v_0 = (v_{max} [B]^n) / (K_A^n + [B]^n)$, and the mean and standard error of the mean (SEM) of the fit parameters for the $N$ measurements are shown. For [B] on the x-axis, total $ParB_F$ concentration was used instead of free $ParB_F$ concentration due to technical issues in estimating the free $ParB_F$ concentration and the meanings of $K_A$ and the cooperativity factor (n) here differ from those in the standard adaptation of the Hill equation. $v_{max}$ is the maximum stimulated $ParA_F$ ATPase turnover rate, $K_A$ is the apparent <u>total</u> concentration of $ParB_F$ necessary for half maximum stimulation, and **n** is the apparent cooperativity coefficient.

| | $ParB_F$ | | $ParB_F$ CTP | | $ParB_F$ CDP | $ParB_F^{R121A}$ | | $ParB_F^{1-42}$ | $ParB_F^{1-42}$ R36A | $ParB_F^{1-42}$-mCherry-EcoRI$^{E111Q}$ | |
| DNA cofactor Number of exp. | Scram N = 6 | $parS_F$ N = 6 | Scram N = 3 | $parS_F$ N = 3 | $parS_F$ N = 3 | Scram N = 3 | $parS_F$ N = 3 | N = 6 | N = 3 | Scram N = 3 | EcoRI DNA N = 2 |
|---|---|---|---|---|---|---|---|---|---|---|---|
| $v_{max}$ (hr$^{-1}$) | 54 ± 5 | 79 ± 2 | 79 ± 8 | 78 ± 8 | 87 ± 5 | 60 ± 5 | 62 ± 5 | 83 ± 3 | 38 ± 2 | 131 ± 13 | 125 ± 9 |
| $K_A$ (µM) | 0.53 ± 0.09 | 0.59 ± 0.03 | 0.41 ± 0.05 | 0.24 ± 0.02 | 0.34 ± 0.04 | 0.86 ± 0.1 | 1.1 ± 0.1 | 1.2 ± 0.1 | 108 ± 13 | 0.16 ± 0.04 | 0.16 ± 0.03 |
| Cooperativity coefficient (n) | 1.2 ± 0.2 | 1.4 ± 0.1 | 3.3 ± 1.0 | 3.6 ± 0.2 | 1.1 ± 0.1 | 2.5 ± 0.6 | 1.6 ± 0.3 | 2.5 ± 0.3 | 1.5 ± 0.2 | 1.4 ± 0.3 | 1.4 ± 0.2 |

interacts with $ParA_F$, but with a much faster dissociation rate constant compared to wild-type $ParB_F$. $ParB_F^{1-42\ R36A}$ could activate $ParA_F$-ATPase with an increased half-saturation concentration of 108 µM, approximately 100-fold higher than $ParB_F^{1-42}$ (*Figure 3D*). These results explain the puzzling report that while the R36A mutation severely compromised activation of $ParA_F$-ATPase by $ParB_F$, it did not impede oscillation of $ParA_F$ on the nucleoid, and only mildly reduced plasmid stability (*Ah-Seng et al., 2013*). At the interface between the $ParA_F$-bound nucleoid and partition complexes containing many $ParB_F$ dimers, the local $ParB_F$ concentration is expected to be sufficiently high for this mutant protein to activate $ParA_F$-ATPase to effectively generate a $ParA_F$ depletion zone and motive force driving the partition complex as indicated by the repeated oscillation of the nucleoid-bound $ParA_F$ distribution.

## $ParA_F$ and $ParB_F$ bind to and dissociate from nsDNA together in the presence of ATPγS with ~ 1:1 stoichiometry

When $ParA_F$-eGFP and full-length $ParB_F$-Alexa647 were incubated together at 1 µM and 2 µM, respectively, in the presence of ATPγS, they bound to the DNA-carpet in parallel maintaining ~1:1 stoichiometry up to a density of ~5000 monomers/µm$^2$ (*Figure 4A*). They also dissociated from the DNA-carpet in parallel, maintaining ~1:1 stoichiometry, when washed with a buffer containing ATPγS, with an apparent dissociation rate constant of approximately ~1 min$^{-1}$ (*Figure 4B*). These results show that $ParA_F$ and $ParB_F$ form a hetero-tetramer containing two monomers each of $ParA_F$ and $ParB_F$ ($A_2B_2$), or larger oligomers composed of the hetero-tetramers, that binds as a unit on nsDNA in the presence of ATPγS.

Full-length $ParB_F$, which forms dimer with apparent $K_D$ of ~ 19 nM (*Figure 4—figure supplement 1*), activated $ParA_F$-ATPase in the presence of nsDNA to ~50 hr$^{-1}$ without significant sigmoidal concentration dependence (*Figure 4E*). Based on the results of experiments with monomeric $ParB_F^{1-42}$ proteins described earlier, we conclude that a single dimer of full-length $ParB_F$ can straddle an nsDNA-bound $ParA_F$ dimer, permitting the two N-termini to interact with both of the $ParB_F$-binding faces of the $ParA_F$ dimer to activate the ATPase.

## In the presence of $parS_F$, $ParB_F$ forms a 2:1 complex with $ParA_F$

We next asked if $ParB_F$ bound to $parS_F$ interacts differently with $ParA_F$ on the DNA-carpet. We preincubated $ParA_F$-eGFP, $ParB_F$-Alexa647, ATPγS and a 24 bp duplex DNA fragment containing a single $parS_F$ consensus sequence, at a slight molar excess over $ParB_F$ dimer, for 10 min at room temperature. At the concentrations used, most of the $ParB_F$ dimers are expected to be bound to $parS_F$. When infused into the DNA-carpeted flow cell, $ParA_F$-eGFP and $ParB_F$-Alexa647 bound to and

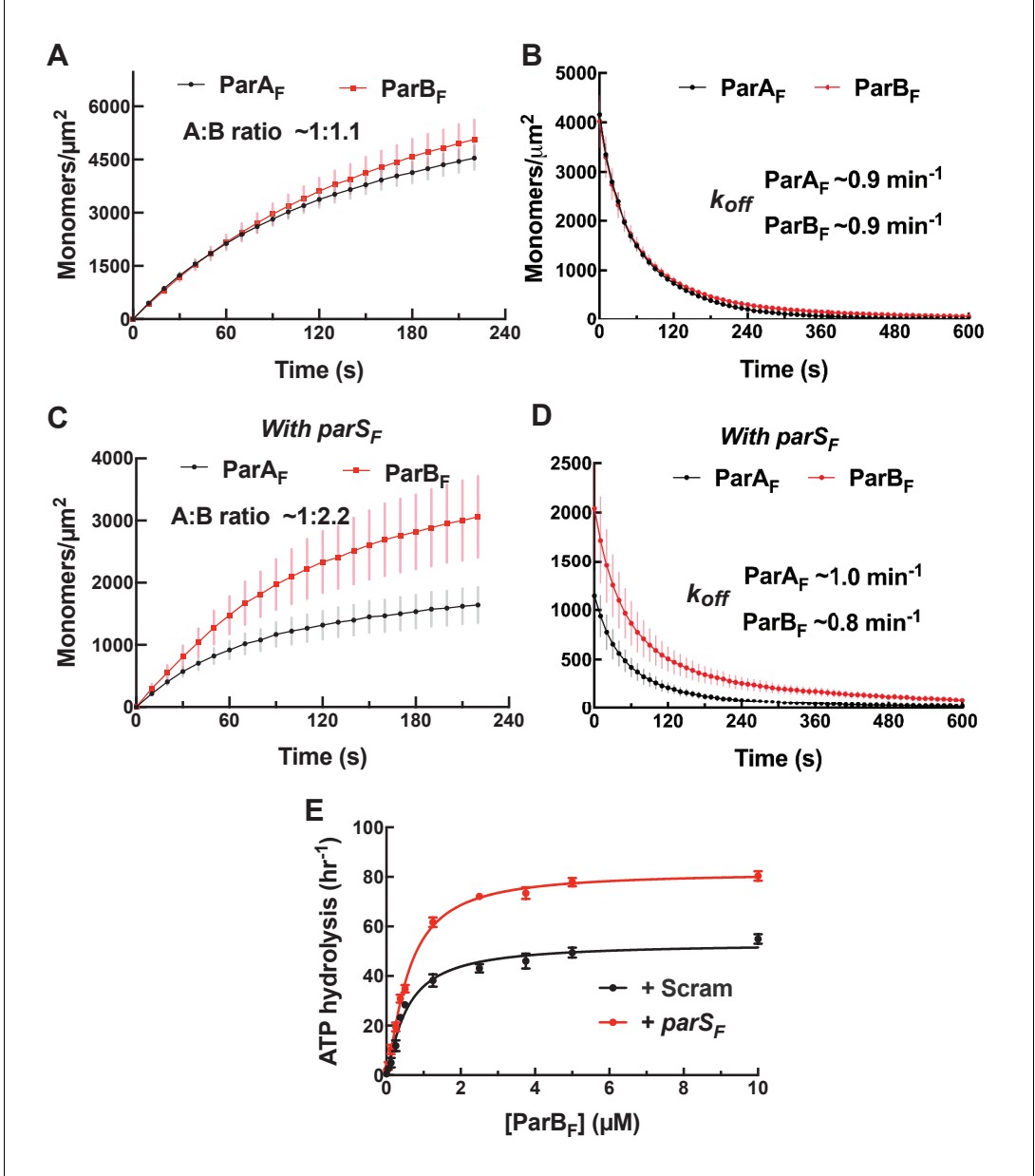

**Figure 4.** $parS_F$ DNA alters protein stoichiometry of the $ParA_F$–$ParB_F$ complex formed prior to ATP hydrolysis and the extent of $ParA_F$-ATPase activation by $ParB_F$. (A) $ParA_F$-eGFP (1 µM) and $ParB_F$-Alexa647 (2 µM) preincubated with ATPγS were infused into the nsDNA-carpeted flow cell and then (B) washed with buffer containing ATPγS. (C, D) As (A) and (B) except the sample included the 24 bp $parS_F$ DNA fragment (1.1 µM). (E) $ParA_F$-ATPase activity was measured in the presence of EcoRI-digested pBR322 DNA (60 µg/ml), different concentrations of $ParB_F$ and either a $parS_F$-DNA fragment or a DNA fragment with a scrambled sequence (1.1-fold higher concentrations than the $ParB_F$ dimers). See *Figure 2* legend and *Tables 1* and *2* for additional details.

The online version of this article includes the following figure supplement(s) for figure 4:

**Figure supplement 1.** Determination of $ParB_F$ monomer-dimer $K_D$ by FRET.

**Figure supplement 2.** Mutation of a BoxII residue R121A does not affect the affinity of $ParB_F^{R121A}$ for $parS_F$, but neither the protein stoichiometry of the $ParA_F$-$ParB_F^{R121A}$ complex assembled on nsDNA prior to ATP hydrolysis, nor the extent of $ParA_F$-ATPase activation by $ParB_F^{R121A}$ is impacted by the presence of $parS_F$.

dissociated from the carpet with a stoichiometry of ~1:2 (*Figure 4C,D*), in sharp contrast to the ~1:1 stoichiometry without $parS_F$ DNA. The kinetic parameters of the complex assembly and disassembly were not significantly affected. These results demonstrate that $ParA_F$ and $ParB_F$ form a complex of one $ParA_F$ dimer and two $ParB_F$ dimers ($A_2B_4$) in the presence of $parS_F$.

Does the change in protein stoichiometry caused by $parS_F$ translate to different levels of $ParA_F$-ATPase activation? A previous study, comparing plasmid DNA with and without $parS_F$ as the cofactor, showed that $ParB_F$ activates $ParA_F$-ATPase a few-fold more efficiently in the presence of plasmid DNA containing a full $parS_F$ site (*Ah-Seng et al., 2009*). We titrated $ParB_F$ in the presence of $ParA_F$, pBR322 DNA, and 24 bp $parS_F$ duplex at a stoichiometric excess concentration over the $ParB_F$ dimer. In the presence of $parS_F$ DNA, $ParB_F$ activated $ParA_F$-ATPase to a maximum turnover rate of ~80 $hr^{-1}$, a ~60% increase compared to reactions where the $parS_F$ fragment was replaced with a scrambled sequence fragment (*Figure 4E*). These results indicate that a single $parS_F$ DNA-bound $ParB_F$ dimer cannot straddle an nsDNA-bound $ParA_F$ dimer to activate the ATPase, but by binding two $ParB_F$ dimers the ATPase activation level reaches slightly higher level than in the absence of $parS_F$ DNA.

We note that $ParB_F^{R121A}$, harboring a mutation in the conserved Box II region of the CTPase domain, neither exhibited a change in the $ParB_F$/$ParA_F$ complex stoichiometry, nor a change in the $ParB_F$-stimulated ATP turnover, in response to $parS_F$ (*Figure 4—figure supplement 2*), suggesting that the effects of $parS_F$ binding described above are mediated through conformational changes in the CTPase domain (see below for further discussion).

## CTP alters the complex formed between $ParA_F$ and $ParB_F$ in a manner similar to $parS_F$ and accelerates complex formation in the presence of $parS_F$

ParB proteins have recently been reported to have CTPase activity that is coupled with changes in their DNA binding properties and refolding of the CTPase domains into a globular dimeric structure in the presence of CTP (*Soh et al., 2019*; *Osorio-Valeriano et al., 2019*) from the more extended and poly-dispersed structure in the absence of nucleotide (*Chen et al., 2015*). We therefore decided to test whether the addition of CTP influences the $ParA_F$–$ParB_F$ complex formed on the DNA-carpet in the presence of ATPγS. When $ParA_F$-eGFP and $ParB_F$-Alexa647 were incubated together in the presence of ATPγS (1 mM) and CTP (2 mM), they bound to and dissociated from the nsDNA-carpet with a stoichiometry of ~1:2 (*Figure 5A*, *Figure 5—figure supplement 1A*). The assembly kinetics of the carpet-bound complex was roughly the same as in the absence of CTP; however, the apparent dissociation rate constant during buffer wash was slightly but reproducibly slower by a factor of roughly two at ~0.6 $min^{-1}$. When $parS_F$ was included together with CTP, the rate of $A_2B_4$ complex assembly on the DNA-carpet increased several-fold, the binding density of the complex on the DNA-carpet reached a correspondingly higher level, and the two proteins dissociated from the DNA-carpet maintaining a ~1:2 stoichiometry with apparent dissociation rate constant similar to that in the absence of $parS_F$ (*Figure 5B*, *Figure 5—figure supplement 1B*). When CTP was replaced by CDP in the presence of $parS_F$, although the $ParB_F$/$ParA_F$ ratio remained above 2, unlike in the presence of CTP, the complex assembly rate did not increase (*Figure 5C*, *Figure 5—figure supplement 1C*), thus behaving similarly to the reaction in the presence of $parS_F$ alone.

Next, we asked if the $parS_F$ DNA fragment was incorporated in the $A_2B_4$ complexes assembled in its presence. The experiments in the presence of $parS_F$ were repeated in the presence or absence of CTP with $ParA_F$ (1 μM), $ParB_F$-Alexa647 (2 μM) and Alexa488-$parS_F$ (1.1 μM), and the nsDNA-carpet-bound ratio of $parS_F$ and $ParB_F$ after 240 s sample infusion was measured (*Figure 5D*). The observed $parS_F$/$(ParB_F)_2$ ratio in the absence of CTP was ~0.2, while in the presence of CTP, the ratio was only ~0.04. Thus, whereas the assembly of the $A_2B_4$ complex involving CTP-$ParB_F$ was accelerated by $parS_F$, a very small fraction of the resulting complex contained the $parS_F$–DNA fragment, indicating that $parS_F$ plays a catalytic role in the activation of CTP-$ParB_F$ and accelerated assembly of the $A_2B_4$ complex. This parallels the observation that a much lower concentration of $parS_F$ fully activated the CTPase activity of $ParB_F$ (*Figure 5—figure supplement 2C*) as has also been shown for $ParB_{Bsu}$ (*Soh et al., 2019*).

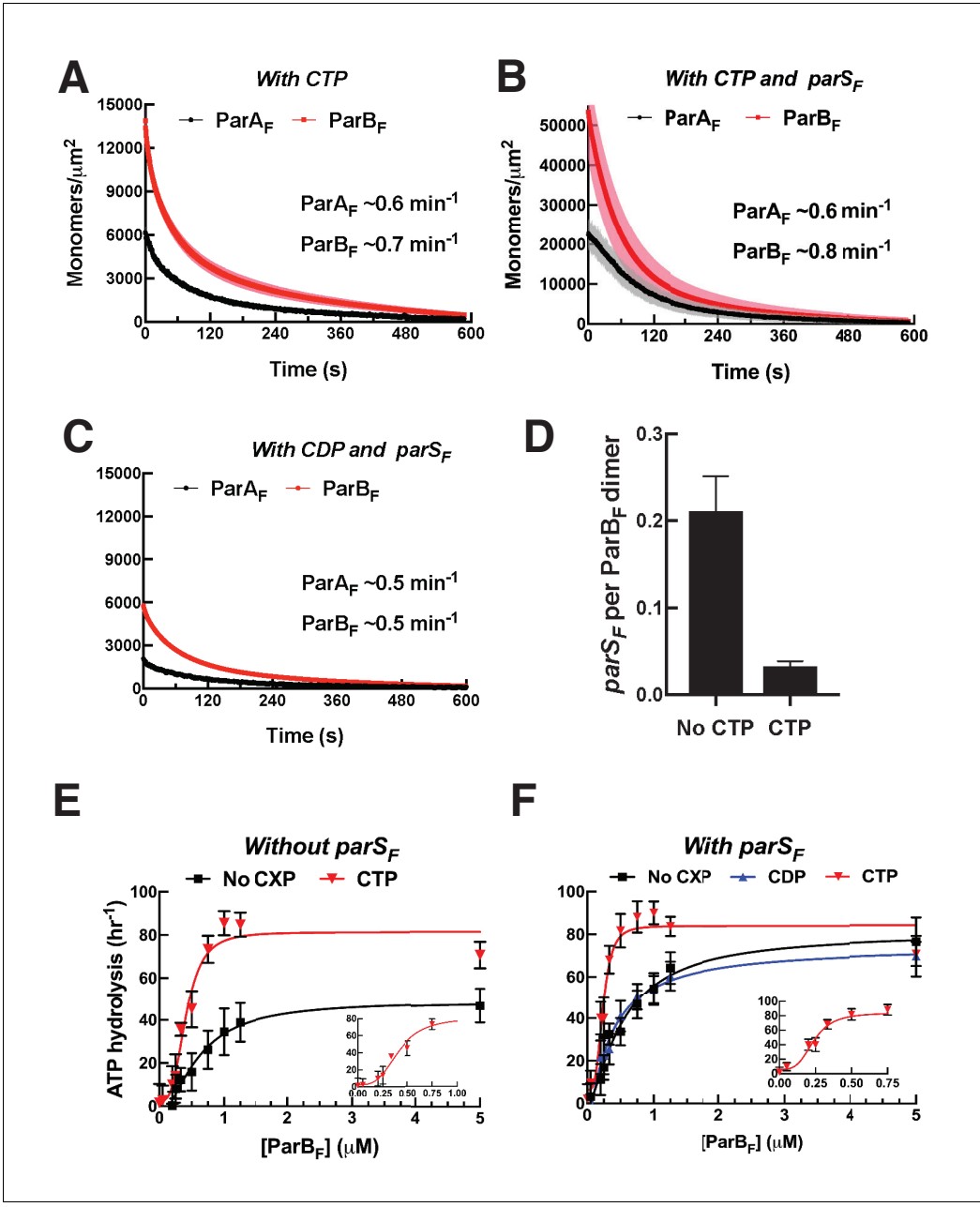

**Figure 5.** CTP and *parS_F* together alter interactions between ParB_F and ParA_F dimers. (**A**) ParA_F–eGFP (1 μM) and ParB_F-Alexa647 (2 μM) preincubated with ATPγS and CTP (2 mM) were infused into the nsDNA-carpeted flow cell and then washed with buffer containing ATPγS and CTP. (**B**) As in (**A**), except a 24 bp *parS_F* fragment (1.1 μM) was added to the sample mixture. (**C**) As in (**B**), except CTP was replaced by CDP. For binding curves, see *Figure 5—figure supplement 1A–C*. (**D**) ParA_F (1 μM), ParB_F-Alexa647 (2 μM) and Alexa488-labeled 24 bp *parS_F* fragment (1.1 μM) preincubated with ATPγS or ATPγS plus CTP (2 mM) were infused into the nsDNA-carpeted flow cell and after 240 s, the ratio of the carpet-bound *parS_F* fragment and ParB_F dimer was measured. (**E**) ParA_F-ATPase activity was measured in the presence of EcoRI-digested pBR322 DNA (60 μg/ml), different concentrations of ParB_F and either no C-nucleotide or 2 mM CTP. Inset shows data in the presence of CTP with expanded abscissa. (**F**) As in (**E**) except the reactions also contained 24 bp *parS_F* fragment (1.1-fold higher concentrations than ParB_F dimers). Inset shows data in the presence of *parS_F* and CTP with expanded abscissa. See *Figure 2* legend and *Tables 1* and *2* for additional details.

The online version of this article includes the following figure supplement(s) for figure 5:

**Figure supplement 1.** Binding curves for ParA_F and ParB_F with CDP or CTP associating with the DNA-carpet.

*Figure 5 continued on next page*

*Figure 5 continued*

**Figure supplement 2.** Nucleotide specificity and CTPase activity of ParB$_F$ in the presence of different concentrations of CTP, in the presence or absence of *parS$_F$*.

## ParB$_F$ activates ParA$_F$-ATPase to the full extent without *parS$_F$* in the presence of CTP

The maximum ParB$_F$ activation of ParA$_F$-ATPase in the presence of CTP, with or without *parS$_F$*, was comparable to that of *parS$_F$*-bound ParB$_F$ in the absence of CTP (*Figure 5E,F*). The half-saturation concentration of ParB$_F$ in the presence of *parS$_F$* and CTP was significantly lower than in the absence of CTP (~0.24 µM vs ~0.6 µM). Combined with the observation of faster assembly of the complex on the DNA-carpet, a likely possibility is that in the presence of CTP and *parS$_F$*, the ParB$_F$ dimer adopts a unique state that interacts with ParA$_F$ dimers with a higher association rate constant. We note that the ParA$_F$-ATPase assays in this study measured radioactive γ-phosphate release from γ-$^{32}$P-ATP, avoiding potential technical complications associated with ATPase measurements in the presence of CTP.

We next measured the ParB$_F$-CTPase activity to estimate the apparent $K_M$ and $k_{cat}$ of ParB$_F$ for CTP hydrolysis in the presence and absence of *parS$_F$*. ParB$_F$ had negligible activity for all NTPs other than CTP (*Figure 5—figure supplement 2A*), and the CTP hydrolysis rate increased with a hyperbolic CTP concentration dependence, which could be fit with the Michaelis–Menten equation with apparent $K_M$ of ~8 µM and ~18 µM and maximum turnover rates of ~14 h$^{-1}$ and ~44 h$^{-1}$ in the absence and presence of *parS$_F$* DNA, respectively (*Figure 5—figure supplement 2B*). Thus, 2 mM CTP used in the experiments of *Figure 5* should have remained saturating ParB$_F$ for the duration of the reaction. Stimulation of the CTPase activity by *parS$_F$* exhibited a pronounced sigmoidal concentration dependence approaching saturation at ~ 60 nM, well below the ParB$_F$ concentration in the reaction (0.84 µM) (*Figure 5—figure supplement 2C*).

During these experiments, which were prompted by a reviewer's comment, we also attempted to characterize the interaction between CDP and ParB$_F$, but discovered that the CDP used here contained ~2% contamination of a compound that released Pi upon incubation with ParB$_F$ (*Figure 5—figure supplement 2D*). The results shown in *Figure 5C* and *Figure 5F* suggest this contamination did not strongly influence the reactions involving CDP, considering that they generally paralleled the results obtained in the absence of C-nucleotides with only minor deviations. However, this contamination prevented us from accurately determining the affinity of ParB$_F$ for CDP.

## In the presence of CTP, ParB$_F$ condenses DNA carrying *parS$_F$* in cis

The recently discovered CTP and *parS*-dependent ParB conformational change appears to promote ParB *parS*-DNA binding and spreading (*Soh et al., 2019*), impacting ParB-DNA partition complex assembly. *In vivo*, spreading ParB forms condensed foci around *parS* sites indicating that *parS*-driven ParB spreading likely occurs *in cis*. Nonetheless, the possibility that *parS* can trigger ParB spreading *in trans* has not been tested *in vitro*. Previous studies reported DNA condensation by *B. subtilis* ParB *via* ParB–ParB interactions, but these studies were conducted in the absence of CTP and did not observe a strong effect of *parS in cis* (*Graham et al., 2014*; *Song et al., 2017*; *Taylor et al., 2015*). To see if *parS$_F$* can act *in trans* and to characterize how *parS$_F$* and CTP influence ParB$_F$–DNA interactions *in vitro*, we conducted single-molecule DNA pulling experiments employing magnetic tweezers. ParB$_F$ at various concentrations was infused into a flow cell containing ~5 kbp DNA tethers that anchored magnetic beads to the coverslip surface (*Figure 6A*). The tethers contained either 12 *parS$_F$* consensus sequence repeats at their midpoints (*parS$_F$*-DNA), or no *parS$_F$* sequence (nsDNA). The protein sample was infused while the DNA tethers were stretched at 5 pN force, preventing DNA condensation. To allow DNA condensation by bound ParB$_F$ molecules, the force was dropped to 0.05 pN and the tether extension was monitored for 30 s. To assess the stability of DNA condensation by ParB$_F$ dimers, tether extension was monitored after increasing the force to 5 pN. In the absence of CTP, we only observed condensation at very high concentrations of ParB$_F$ (>5 µM) and did not see a significant difference between *parS$_F$*-containing and non-specific tethers (*Figure 6B* inset). However, in the presence of CTP, 50 nM ParB$_F$ robustly condensed *parS$_F$*-containing DNA tethers (*Figure 6B* purple). These condensed protein-DNA complexes resisted 5 pN extension force,

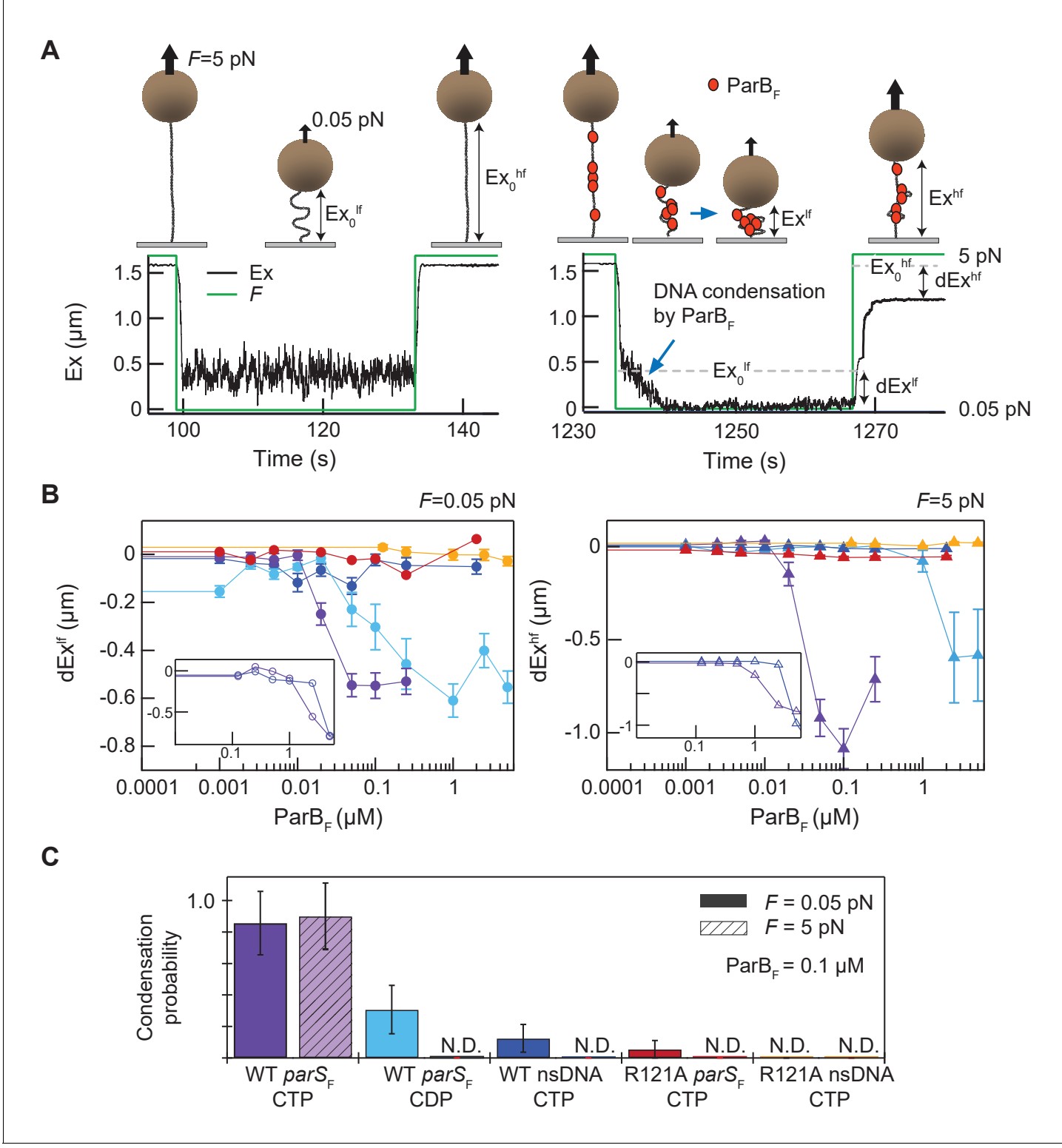

**Figure 6.** Magnetic tweezers measurements of $parS_F$ and CTP-dependent DNA condensation by $ParB_F$. (**A**) Schematic showing the magnetic tweezers DNA condensation assay. One end of a 5 kb DNA molecule is attached to the surface of a flow-cell and the free end is attached to a 1 μm magnetic bead (brown sphere). The DNA extension (Ex) was measured by tracking the bead height above the cover glass surface at two different forces; 0.05 pN (low force, lf), and 5 pN (high force, hf). The extent of DNA condensation was estimated from the difference in DNA extension with and without $ParB_F$. (**B**) Changes in extension at low force ($dEx^{lf} = Ex^{lf} - Ex_0^{lf}$), left panel, and at high force ($dEx^{hf} = Ex^{hf} - Ex_0^{hf}$), right panel, for seven different conditions plotted as a function of $ParB_F$ concentration. The extension values were the averages of the last 5 s of the extension at low force (circles) and the first 5

*Figure 6 continued on next page*

*Figure 6 continued*

s of the extension at high force (triangles). Error bars represent standard error of means (SEM). Different conditions are color coded as follows. Purple: *parS_F*-DNA tether with WT ParB_F and CTP; light blue: *parS_F*-DNA tether with WT ParB_F and CDP; dark blue: nsDNA tether with WT ParB_F and CTP; red: *parS_F*-DNA tether with ParB_F^R121A and CTP; orange: nsDNA tether with ParB_F^R121A and CTP. For comparison with condensation in the presence of CTP, dEx data of *parS_F*-DNA tether (purple) and nsDNA tether (blue) with WT ParB_F without CTP are displayed (inset, open circles for 0.05 pN, triangle for 5 pN respectively). (C) The condensation probabilities at 0.1 μM ParB_F for five different conditions at 0.05 pN and 5 pN. The condensation probability was calculated by dividing the number of DNA tethers that exhibited DNA condensation by the total number of DNA tethers for each measurement condition. Except for *parS_F*-DNA with WT ParB_F and CTP, all conditions show either minimal or negligible condensation probabilities. The different conditions are color-coded as indicated in (B), and the diagonal stripes indicates probabilities at 5 pN. Error bars represent standard error of means (SEM).

The online version of this article includes the following figure supplement(s) for figure 6:

**Figure supplement 1.** Stepwise de-condensation of condensed DNA tethers in the presence of CDP or CTP by tensile force: condensation observed in the presence of CDP is unstable.

**Figure supplement 2.** Tether condensation by ParB_F is comparable for topologically constrained (supercoilable) and unconstrained (nicked) DNA.

**Figure supplement 3.** DNA tethers without *parS_F* sequence are not condensed by ParB_F and *parS_F in trans* in the presence of CTP.

requiring many minutes at 5 pN tension to de-condense (*Figure 6—figure supplement 1A*). The slow de-condensation took place through a series of abrupt steps, which we interpret as stepwise opening of large DNA loops held by multiple ParB_F–ParB_F interactions (*Figure 6—figure supplement 1A*). Condensation was comparable for DNA molecules that were topologically constrained, i.e., could be supercoiled, or unconstrained (nicked), suggesting that condensation is not a consequence of topological changes in the DNA caused by ParB_F translocating away from *parS_F* sites (*Figure 6—figure supplement 2*). We observed some condensation events with *parS_F* containing tethers in the presence of CDP, but these events were rarer, required higher ParB_F concentrations, and were almost completely de-condensed within 5 s of raising the force to 5 pN, in stark contrast to condensation in the presence of CTP (*Figure 6B*, light blue, *Figure 6—figure supplement 1B*). Since this experiment was also carried out using CDP that contained a compound hydrolysable by ParB_F, contribution of this compound to the limited tether condensation cannot be ruled out. In contrast, ParB_F was unable to substantially condense DNA tethers lacking *parS_F* sequences, even in the presence of CTP, and rare condensation events were quickly reversed by the application of 5 pN force (*Figure 6B*, dark blue). Addition of *parS_F*-containing DNA fragments together with ParB_F and CTP did not rescue the inability to condense tethers lacking *parS_F*, indicating that *parS_F* cannot act *in trans* to promote ParB spreading and condensation of DNA molecules (*Figure 6—figure supplement 3*). Together these results indicate that *parS_F* mediates loading of multiple CTP-bound ParB_F dimers *in cis* onto the DNA-tethers and these ParB_F dimers are capable of forming DNA looping bridges likely *via* inter-dimer interactions to form a condensed partition complex-like structure. As expected, ParB_F^R121A bearing a mutation at the critical Box II residue in the CTPase domain was unable to condense DNA to a significant degree even with *parS_F* containing tethers (*Figure 6B*, red). We propose that stable DNA condensation by ParB_F is mediated by CTPase domain dimerization and requires both *parS_F* and CTP at moderate ParB_F concentrations (~100 nM) (*Figure 6C*).

## Discussion

In this report, we characterized facets of the ParA_F–ParB_F interaction leading to the assembly of the nsDNA-bound ParA_F–ParB_F complex that is required to activate ParA_F for ATP hydrolysis and dissociation from nsDNA under the influences of *parS_F* and CTP (summarized in *Figure 7*). Our results indicate that both ParB_F binding faces of the nsDNA-bound ParA_F dimers must be occupied by a ParB_F N-terminal domain for ATPase activation (*Figures 2C–G* and *7B*). In principle, two copies of the ParB_F N-terminal domain activating a ParA_F dimer could belong to one ParB_F dimer as seen in the absence of CTP or *parS_F* (*Figures 4A,B* and *7B*, middle). However, most ParB_F dimers in partition complexes *in vivo* are likely in the CTP- and *parS_F*-activated state, spreading over a *parS_F*-proximal DNA region. CTP or *parS_F* binding alters the ParB_F dimer structure to prevent a single ParB_F dimer from providing both copies of the N-terminal domain to occupy both binding faces of a ParA_F dimer, necessitating two ParB_F dimers, each providing one N-terminal domain to a ParA_F dimer (*Figures 4C,D*, *5*, and *7B*, bottom). Strikingly, *parS_F* together with CTP significantly increased the

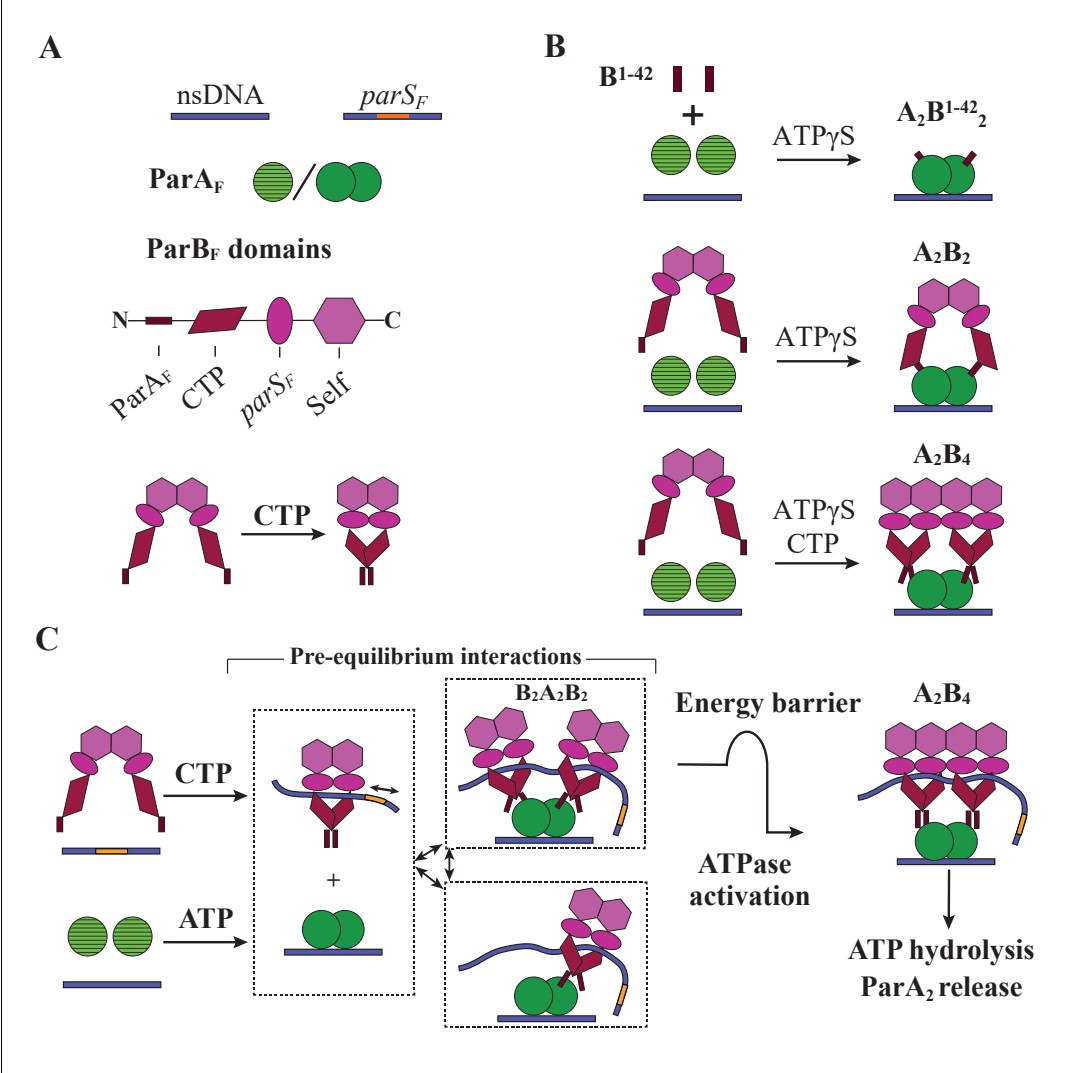

**Figure 7.** Cartoon of the proposed pre-ATPase-activation complexes of $ParA_F$ and $ParB_F$. (A) Pictograms of nsDNA, $parS_F$-DNA, $ParA_F$ monomer/dimer and $ParB_F$ domains with binding ligand designations. The CTPase domains of a $ParB_F$ dimer fold forming a single globular domain on binding CTP, bringing the two $ParA_F$-binding domains into close proximity. (B) $ParA_F$-binding domain, $ParB_F^{1-42}$ alone can convert $ParA_F$ monomers to DNA-binding-competent dimers in the presence of ATPγS by forming an $A_2B^{1-42}_2$ complex (top). $ParB_F$ dimers in the absence of CTP convert $ParA_F$ monomers to DNA-binding-competent dimers in the presence of ATPγS by straddling a $ParA_F$ dimer to form an $A_2B_2$ complex (middle). In the presence of CTP, the close proximity of the $ParA_F$-binding domains of the $ParB_F$ dimer prevents $A_2B_2$ complex formation and instead an $A_2B_4$ complex assembles on nsDNA in the presence of ATPγS (bottom). (C) In the presence of $parS_F$ and CTP, $ParB_F$ dimers load on to the $parS_F$–DNA and spread to adjacent DNA regions while adopting a state that enables faster assembly of $A_2B_4$ complexes. Considering the requirements for efficient partition complex motion by diffusion-ratchet mechanism based on the chemophoretic principle of force generation, we propose a significant energy barrier that slows the formation of the ATP hydrolysis-competent $A_2B_4$ complex. This energy barrier partially decouples $ParA_F$–$ParB_F$ association–dissociation dynamics from ATP hydrolysis, which triggers $ParA_F$ dissociation from the nucleoid.

$A_2B_4$ complex assembly rate on nsDNA without strongly affecting its disassembly rate. Although we have not analyzed the full kinetic details of the process that leads to ATPase activation by $ParB_F$, we propose that a moderately slow transition separates formation of the ATPase-activated $A_2B_4$ complex from the rapidly reversible $ParA_F$–$ParB_F$ interaction processes. Such a local slow step would partially decouple the reversible $ParA_F$–$ParB_F$ interaction dynamics from the irreversible ATP hydrolysis, thereby promoting dynamic interactions between the nucleoid and partition complex that facilitate partitioning *via* the diffusion-ratchet mechanism as elaborated below.

The clearest indication that both $ParB_F$-interacting faces of the nsDNA-bound $ParA_F$ dimer must be occupied by the N-terminal domain of $ParB_F$ for ATPase activation came from experiments using

artificial ParB$_F$ constructs. We showed that monomeric ParB$_F^{1-42}$ stimulates ParA$_F$ ATPase with a clear sigmoidal concentration dependence, indicating that one molecule of ParB$_F^{1-42}$ binding to one side of a ParA$_F$ dimer cannot fully activate the ParA$_F$ ATPase (*Figure 2F*). When ATP hydrolysis was blocked by using non-hydrolysable ATPγS, ParB$_F^{1-42}$-mCherry formed an equimolar complex with ParA$_F$ (*Figure 2C–E*). Thus, the ParA$_F$ forms a complex with ParB$_F^{1-42}$-mCherry bound at both ParB$_F$-interacting faces of the ParA$_F$ dimer prior to ATP hydrolysis (*Figure 7B*; top). Consistently, an artificially dimeric ParB$_F^{1-42}$ construct, ParB$_F^{1-42}$-mCherry-EcoRI$^{E111Q}$, efficiently activated ParA$_F$-ATPase with hyperbolic concentration dependence (*Figure 2G*).

In the absence of CTP or *parS$_F$*, the ParB$_F$ dimer is held together by the C-terminal self-dimerization domain (*Figure 7A*, bottom) with dimerization $K_D$ of ~19 nM (*Figure 4—figure supplement 1*). In this state, the N-terminal halves of the monomers are thought to be separate from each other according to the SAXS envelope of the structure (*Chen et al., 2015*; also see *Figure 2—figure supplement 3B*). Thus, one dimer could straddle a ParA$_F$ dimer with each N-terminal ParA$_F$-interaction domain interacting with one of the two faces of a ParA$_F$ dimer (*Figure 7B*, middle).

In contrast, when bound by CTP the CTPase domains in a ParB$_F$ dimer fold to form a single globular domain (*Figure 7A*, bottom) (*Soh et al., 2019*; see *Figure 2—figure supplement 3C*). The two ParA$_F$-interaction domains emanating from this dimeric domain are unlikely to reach both sides of a ParA$_F$ dimer, necessitating an A$_2$B$_4$ complex for ATPase activation (*Figure 7B*, bottom). In theory, it is possible that a chain of (A$_2$B$_2$)$_n$ might form, but the 1:2 protein stoichiometry observed in the presence of ATPγS indicates such a configuration is unfavored. A$_2$B$_4$ complexes formed in the presence of CTP and *parS$_F$* DNA fragments contained almost no *parS$_F$* DNA fragments (*Figure 5D*). This is consistent with the notion that after binding *parS$_F$*, CTP-ParB$_F$ dimers convert to a low *parS$_F$*-affinity state while remaining topologically bound to the DNA and spreading to surrounding DNA regions (*Soh et al., 2019*). The A$_2$B$_4$ complex formed in the presence of *parS$_F$* DNA fragments, even without CTP, contained significantly less than a stoichiometric amount of *parS$_F$* fragment (*Figure 5D*). This suggests that association with nsDNA-bound ParA$_F$ dimer lowers the affinity of ParB$_F$ for *parS$_F$*, perhaps shifting the structure closer toward *parS$_F$*-activated ParB$_F$-CTP.

The ParB:ParA stoichiometry change from 1:1 to 2:1 caused by *parS$_F$* (*Figure 4C,D*) did not occur with the Box II mutant ParB$_F^{R121A}$ (*Figure 4—figure supplement 2*). It is possible that when a ParB$_F$ dimer binds *parS$_F$*, the adjacent CTPase domains of the two monomers adopt a mutually interacting folded state akin to the CTP-bound state even without CTP, promoting a Box II dependent dimerized domain structure. This may disfavor formation of the A$_2$B$_2$ complex, favoring the A$_2$B$_4$ complex that was observed.

All the A$_2$B$_2$ and A$_2$B$_4$ complexes we observed in the presence of ATPγS dissociated from nsDNA more slowly (k$_{off}$ = 0.5–1 min$^{-1}$; *Source data 1*) compared to ParA$_F$ in the absence of ParB$_F$ (~6 min$^{-1}$; *Figure 2B*). For the case of monomeric ParB$_F^{1-42}$, which dissociated from ParA$_F$ more rapidly, the presence of 10 µM ParB$_F^{1-42}$ in the wash buffer restored the low apparent nsDNA dissociation rate constant of the complex (*Figure 2E*). Nevertheless, ParA$_F$ dimers in these complexes appear to be primed for further conformational change toward less stably DNA-associated state. Upon dissociation of ParB$_F^{1-42}$ from the A$_2$B$^{1-42}_2$ complex, ParA$_F$ dissociated from the DNA-carpet within a second or so, much faster than the ATPγS-ParA$_F$-dimer that has not yet formed a A$_2$B$_2$ or A$_2$B$_4$ complex (*Figures 2D* and *3B*).

Our single-molecule DNA condensation measurements indicate that CTP-bound ParB$_F$ dimers are activated by contacting *parS$_F$* to load *in cis* onto the *parS$_F$*-carrying DNA in numbers exceeding the copy number of the *parS$_F$*-consensus sequence (ParB spreading) as shown by others for chromosomal ParBs (*Jalal et al., 2020*; *Soh et al., 2019*), and condense the DNA forming an *in vivo* partition complex-like structure (*Figure 6*). Although the magnetic tweezers instrument used in this study did not allow direct measurement of the number of ParB$_F$ molecules contained in the condensed DNA, the large number of de-condensation steps observed when high tension was applied is consistent with the presence of a large number of ParB$_F$ dimers in the condensed DNA (*Figure 6*, *Figure 6—figure supplement 1A*). CDP failed to support efficient condensation of *parS$_F$*-carrying DNA by ParB$_F$ and the limited condensation observed, which could be due to the contaminating material in the CDP used, was disrupted far more readily than CTP-supported condensates (*Figure 6*, *Figure 6—figure supplement 1B*). Our results show that DNA-condensation is caused by ParB$_F$–ParB$_F$ interactions forming DNA-looping bridges without requiring other protein factors. Combined with evidence indicating that *parS$_F$*-activated ParB$_F$–CTP adopts a unique conformational state

(*Soh et al., 2019*), we favor the view that DNA-bridging capability, mediated by inter-dimer ParB$_F$ interaction, is another attribute of this ParB$_F$ state.

Our observation indicates that the state of ParB$_F$ discussed above is maintained after release from *parS$_F$*-containing DNA. ParB$_F$ associates with nsDNA-bound ParA$_F$ dimers forming the A$_2$B$_4$ complex with a faster apparent assembly rate in the presence of *parS$_F$* and CTP together than with either CTP or *parS$_F$* alone. This observation is consistent with the decreased half-saturation concentration in the ATPase activation assay (*Figure 5F*, *Table 2*). According to the sliding clamp model of spreading ParB–CTP dimers proposed by *Soh et al., 2019*, ParB$_F$–CTP dimers loaded onto a short *parS$_F$* DNA fragments would quickly slide off the DNA as shown by *Jalal et al., 2020*. Since our ATPase activation assay and the DNA-carpet-bound A$_2$B$_4$ complex assay in the presence of CTP and *parS$_F$* were done using a short linear *parS$_F$* DNA fragment, the *parS$_F$*-activated state of the ParB$_F$–CTP dimers we described in this study must remain in this 'activated' state for an extended period after sliding off the *parS$_F$* fragment. Accordingly, the A$_2$B$_4$ complexes bound to the DNA-carpet in the presence of CTP, ATPγS and *parS$_F$* fragments contained almost no *parS$_F$* fragments (*Figure 5D*). Thus, *parS$_F$* acts as a catalyst to convert ParB$_F$–CTP dimers from a pre-activation state to an activated state capable of faster A$_2$B$_4$ complex assembly. This notion is also consistent with the observation that significantly less than a stoichiometric concentration of *parS* DNA relative to ParB is sufficient for full activation of the ParB CTPase (*Figure 5—figure supplement 2C*; *Soh et al., 2019*). Although the ParB$_F$ dimers in this activated state failed to load efficiently onto DNA lacking *parS$_F$* sequences *in trans* (*Figure 6—figure supplement 3*), in the absence of contrary evidence, the parsimonious assumption is that this ParB$_F$ dimer retains the conformation of spreading ParB$_F$ dimers that remain loaded on the *parS$_F$*-containing DNA *in cis*. Thus, we propose that the functional properties of ParB$_F$ we observed in the presence of CTP and *parS$_F$*, both in facilitating assembly of A$_2$B$_4$ complexes and in activating ParA$_F$-ATPase, reflect those of the majority of ParB$_F$ dimers in partition complexes *in vivo*.

Our study, together with previous studies, indicates that the ATP turnover rates of ParABS systems are slow because of multiple, slow kinetic steps. These slow steps are strategically placed in the reaction pathway in order to tune the system and drive the motion of the partition complex through the diffusion-ratchet mechanism (*Sugawara and Kaneko, 2011*; *Vecchiarelli et al., 2010*). Even at saturating concentrations of ParB$_F$ in the *parS$_F$*-activated CTP-bound state, the maximum ATP turnover rate of ParA$_F$ remained modest (~80 ATP/ParA$_F$-monomer/hour; *Figure 5*). The slow reactivation of ParA nucleoid binding after ATP hydrolysis likely dominates the overall ATPase cycle time (*Vecchiarelli et al., 2010*). The presence of a large fraction of ParA$_F$ in the DNA-unbound state during the steady-state ATPase assay was evidenced by the fact that the half-saturation concentration of ParB$_F$ (in the presence of CTP and *parS$_F$*) forming the A$_2$B$_4$ complex was ~0.2 μM, while the total ParA$_F$ concentration was 1 μM, suggesting less than ~20% of ParA$_F$ was in the nsDNA-bound state ready to interact with ParB$_F$. *In vivo* the reactivation rate is likely lower since nucleoid-bound ParA is only fully activated on encountering the partition complex, which lowers the concentration of ParA in the cytosol waiting to be reactivated. The lower precursor concentration slows the nucleoid rebinding rate of ParA non-linearly because reactivation involves a relatively fast nucleotide-dependent reversible ParA dimerization with apparent $K_D$ of ~2 μM, followed by a slow conformational step. This makes the process dimerization-limited at lower precursor ParA concentrations according to the study of ParA$_{P1}$ (*Vecchiarelli et al., 2010*). Whereas this slow ParA reactivation and rebinding process, which allows the maintenance of the nucleoid-bound ParA concentration gradient (*Hu et al., 2017*), is a critical element of chemophoresis driven motility, the rate of ParA-ATPase activation by ParB is another important factor. In particular, efficient chemophoresis force generation relies on ParA$_F$–ParB$_F$ interactions achieving a local quasi-equilibrium prior to ATP hydrolysis (*Sugawara and Kaneko, 2011*). Therefore, we speculate that there is a significant energy barrier associated with the conformational transition of a ParA$_F$–ParB$_F$ complex to achieve ATPase activation (*Figure 7C*). The resulting local time delay, in addition to the fact that two ParB dimers are required to bind a ParA dimer to activate its ATPase, would partially decouple the pre-ATP hydrolysis ParA$_F$–ParB$_F$ reversible interaction steps from the ATP hydrolysis step. This delay would in turn permit ParB–ParA binding to approach local quasi-equilibrium, increasing the efficiency of ParA distribution gradient sensing and motive force generation by the partition complex. In addition, this slow activation step would prevent possible over-depletion of the local nucleoid-bound ParA$_F$ as the partition complex establishes the ParA$_F$ depletion zone.

Disassembly of the ATP-bound $A_2B_4$ complex might be slow prior to ATP hydrolysis considering the stability of the complexes in the presence of ATPγS. Thus, we propose the energy barrier postulated above is positioned immediately prior to formation of this complex rather than between this complex assembly and ATP hydrolysis. A slow step after formation of the stable complex would prolong the lifetime of the link between the nucleoid and the partition complex impeding partition complex motion without permitting the reversible $ParA_F$–$ParB_F$ interaction to approach equilibrium. We consider this conformational transition is likely the step synergistically accelerated by CTP and $parS_F$. We note that CTP-activated $ParB_F$ stimulates $ParA_F$ ATPase with sigmoidal concentration dependence (*Figure 5E,F*, *Table 2*), suggesting two $ParB_F$ dimers separately bind a $ParA_F$ dimer during a pre-equilibrium binding phase, forming a transient $B_2A_2B_2$ complex. We imagine the slow conformational step proposed here might be assisted by the property of the CTP/$parS_F$-activated $ParB_F$ dimers that promotes inter-dimer interactions as suggested by the magnetic-tweezers experiments, stabilizing the interaction between the two $ParB_F$ dimers within a complex, depicted as conversion of $B_2A_2B_2$ complex to $A_2B_4$ complex in *Figure 7C*. This might explain the higher assembly rate and stability of the complex formed with $parS_F$-activated $ParB_F$-CTP. Yet, CTP and $parS_F$ DNA do not significantly increase the ATP turnover rate of ~80 $h^{-1}$, indicating that the proposed kinetic delay time must be a small fraction of the ATPase cycle time (~45 s), for which we believe the rate limiting step resides in the reactivation process of ParA for nsDNA binding after ATP hydrolysis (*Vecchiarelli et al., 2010*). Assembly of the $A_2B^{1-42}_2$ complex perhaps does not experience this time delay due to fewer steric constraints, but $ParB_F^{1-42}$ dissociates more readily compared to full-length $ParB_F$. If two CTP-bound and $parS_F$-activated $ParB_F$ dimers independently associating with a nucleoid-bound $ParA_F$-ATP dimer is important for efficient partition complex motive force generation by the chemophoretic principle as proposed above, one might be able to design a mutant $ParB_F$ that can activate $ParA_F$-ATPase by forming an $A_2B_2$ complex even in the presence of CTP, which would significantly affect plasmid partition efficiency. Efforts to generate such $ParB_F$ mutants are currently under way.

This study demonstrates how $parS_F$, along with CTP, has wide-reaching roles in the F-plasmid ParABS system; not only in $ParB_F$'s ability to spread from $parS_F$ and promote $ParB_F$–$ParB_F$ interactions for partition complex compaction, but also in $ParB_F$ dimer interactions with $ParA_F$. However, we still need to investigate how the $ParB_F$ CTPase activity is impacted by $parS_F$ in different states of the $ParB_F$–$parS_F$ complex and its interaction with the $ParA_F$-DNA complex. More generally, in order to understand how the system is orchestrated to achieve system dynamics that result in robust plasmid segregation, improved understanding of the microscopic kinetic parameters is essential. Many details of the system dynamics still remain to be addressed to understand the full picture of the ParABS partition mechanism.

# Materials and methods

## Key resources table

| Reagent type (species) or resource | Designation | Source or reference | Identifiers | Additional information |
|---|---|---|---|---|
| Strain, strain background (*Escherichia coli*) | BL21 DE3 AI | Invitrogen | C607003 | Protein expression strain |
| Recombinant DNA reagent | pET11a | EMD Millipore | 9436 | Protein expression vector |
| Recombinant DNA reagent | pET28a-*parS_F* | This work | | Tether DNA PCR template |
| Recombinant DNA reagent | pET28a | EMD Millipore | 69865 | Tether DNA PCR template |
| Recombinant DNA reagent | pBlueScript II KS(+) | Agilent | 212207 | Tether DNA PCR template |

*Continued on next page*

*Continued*

| Reagent type (species) or resource | Designation | Source or reference | Identifiers | Additional information |
|---|---|---|---|---|
| Recombinant DNA reagent | pX7 | *Vecchiarelli et al., 2013* | | $ParA_F$ overexpression plasmid |
| Recombinant DNA reagent | pX2 | *Vecchiarelli et al., 2013* | | $ParA_F$-eGFP overexpression plasmid |
| Recombinant DNA reagent | pX8 | *Vecchiarelli et al., 2013* | | $ParB_F$ overexpression plasmid |
| Recombinant DNA reagent | pET11a-ParB$_F^{R121A}$ | This work | | $ParB_F^{R121}$ overexpression plasmid |
| Recombinant DNA reagent | pET11a-ParB$_F^{1-42}$-mCherry | This work | | $ParB_F^{1-42}$-mCherry overexpression plasmid |
| Recombinant DNA reagent | pET11a-ParB$_F^{1-42\ R36A}$-mCherry | This work | | $ParB_F^{1-42\ R36A}$-mCherry overexpression plasmid |
| Recombinant DNA reagent | pET11a-ParB$_F^{1-42}$-mCherry-EcoRI$^{E111Q}$ | This work | | $ParB_F^{1-42}$-mCherry-EcoRI$^{E111Q}$ overexpression plasmid |
| Sequence-based reagent | *parS$_F$* DNA | This work | | 5'-AGTCTGGGACCA CGGTCCCACTCG |
| Sequence-based reagent | *parS$_F$* DNA Alexa 488 | This work | | 5'-Alexa488-(HNS)-AGTCTGGGACCAC GGTCCCACTCG |
| Sequence-based reagent | *parS$_F$* DNA complement strand | This work | | 5'-CGAGTGGGACC GTGGTCCCAGACT |
| Sequence-based reagent | Scrambled seq DNA | This work | | 5'-AGTCTGCAGCTAC TATACCACTCG |
| Sequence-based reagent | Scrambled seq DNA complement strand | This work | | 5'-CGAGTGGTATAGT AGCTGCAGACT |
| Sequence-based reagent | EcoR1 sequence + strand | This work | | 5'-GAATTCCGAGTGGG ACCGTGGTCCCAGTCT GATTATCAGACCGAGA ATTCAAGTTGGGACC GTGGTCCCAAGAGAAT |
| Sequence-based reagent | EcoR1 sequence - strand | This work | | 5'-ATTCTCTTGGGACCAC GGTCCCAACTTGAATTC TCGGTCTGATAATCAGA CTGGGACCACGGTCCC ACTCGGAATTC |
| Sequence-based reagent | 5 kb DNA primer1 | *Seol and Neuman, 2011* | | 5'- GCTGGGTCTCGGTT GTTCCCTTTAGTGAG GGTTAATTG |
| Sequence-based reagent | 5 kb DNA primer2 | *Seol and Neuman, 2011* | | 5'- GCTGGGTCTCGTG GTTTCCCTTTAGTG AGGGTTAATTG |
| Sequence-based reagent | DNA handle primer1 | *Seol and Neuman, 2011* | | 5'- GGACCTGCTTTCG TTGTGGCGTAATC ATGGTCATAG |
| Sequence-based reagent | DNA handle primer2 | *Seol and Neuman, 2011* | | 5'- GGGTCTCGTGG TTTATAGTCCTG TCGGGTTTC |
| Peptide, recombinant protein | ParB$_F^{1-42}$ | This work | | MKRAPVIPKHTLNT QPVEDTSLSTPAAP MVDSLIARVGVMAR |
| Peptide, recombinant protein | ParB$_F^{1-42\ R36A}$ | This work | | MKRAPVIPKHTLNTQ PVEDTSLSTPAAPM VDSLIAAVGVMAR |

*Continued on next page*

*Continued*

| Reagent type (species) or resource | Designation | Source or reference | Identifiers | Additional information |
|---|---|---|---|---|
| Chemical compound, drug | ATP | Millipore-Sigma | A2383 | |
| Chemical compound, drug | GTP | Millipore-Sigma | G8877 | |
| Chemical compound, drug | UTP | Thermo Scientific | J23160 | |
| Chemical compound, drug | CTP | Millipore-Sigma | C1506 | |
| Chemical compound, drug | CDP | Millipore-Sigma | C9755 | 2–3% possible contamination of ParB$_F$-CTPase substrate detected |
| Chemical compound, drug | $\gamma^{32}$P-ATP | Perkin-Elmer | NEG002A250UC | |
| Chemical compound, drug | Dynabeads MyOne Streptavidin T1 | Invitrogen | 65601 | |
| Chemical compound, drug | Alexa Fluor 488 C5 Maleimide | Thermo Fisher | A10254 | |
| Chemical compound, drug | Alexa Fluor 594 C5 Maleimide | Thermo Fisher | A10256 | |
| Chemical compound, drug | Alexa Fluor 647 C2 Maleimide | Thermo Fisher | A20347 | |
| Chemical compound, drug | Antifoam Y-40 emulsion | Sigma | A5758 | |
| Chemical compound, drug | EDTA-free Sigmafast protease inhibitor cocktail tablet | Sigma | S8830 | |
| Chemical compound, drug | DOPC | Avanti polar lipids | 850375C | |
| Chemical compound, drug | DOPE-Biotin | Avanti polar lipids | 850149P | |
| Chemical compound, drug | Biotin-14-dCTP | Thermo Fisher | 19518018 | |
| Chemical compound, drug | Biotin-16-dUTP | Roche | 11093070910 | |
| Chemical compound, drug | Digoxigenin-11-dUTP | Roche | 11093088910 | |
| Commercial assay or kit | EnzChek Phosphate assay kit | Thermo Fisher | E6646 | |

*Continued on next page*

*Continued*

| Reagent type (species) or resource | Designation | Source or reference | Identifiers | Additional information |
|---|---|---|---|---|
| Software, algorithm | Prism 8 | GraphPad | Prism 8 | Used for curve fitting, and fitting parameters and their error estimation. |
| Software, algorithm | Igro Pro 7 | Wavemetrics | Igro Pro | Used for single molecule data analysis. |
| Software, algorithm | LabVIEW | National Instruments | LabView NXG Full | Used for instrumental control in single molecule experiments. |
| Software, algorithm | Metamorph 7 | Molecular Devices | Metamorph 7 | Used for TIRF michroscope data acquisition. |
| Software, algorithm | ImageJ/Fiji | National Institutes of Health | ImageJ | Used for TIRF michroscope image analysis. |
| Other (Instrument) | Prism type TIRF microscope | In house *Ivanov and Mizuuchi, 2010*; *Vecchiarelli et al., 2013* | | Used for ParAF-ParBF complex assembly-disassembly experiments. |
| Other (Instrument) | Magnetic tweezers | In house *Seol and Neuman, 2011* and *Seol et al., 2016* | | Used for taking single molecule measurements of enzyme binding on 5 kb DNA. |
| Other (Instrument) | Plate reader | BMG Labtech | Clariostar Plus | Used for FRET-based ParBF dimerization $K_D$ and CTP hydrolysis assays using EnzChek Phosphate assay kit |

## Plasmids and constructs for protein expression

All expression open-reading frames were synthesized and subcloned into pET11a (Genscript). $ParA_F$, $ParA_F$-eGFP, $ParB_F$, $ParB_F^{R121}$, $ParB_F^{1-42}$-mCherry, $ParB_F^{1-42\ R36A}$-mCherry, and $ParB_F^{1-42}$-mCherry-EcoRI$^{E111Q}$ constructs were made with a hexa-histidine tag on their C-terminus. Protein fusions were made with a SGGG linker between fused domains, with exception of $ParB_F^{1-42}$-mCherry-EcoRI$^{E111Q}$, which had a $4\times$ (SGGG) linker between $ParB_F^{1-42}$ and mCherry. $ParB_F^{1-42}$ and $ParB_F^{1-42\ R36A}$ were synthesized *de novo* (Genscript).

## Oligonucleotides

The 24 bp double-stranded DNA fragments containing the $parS_F$ consensus sequence and a scrambled sequence used in this study were as follows: 5′-AGT CTG GGA CCA CGG TCC CAC TCG; 5′-AGT CTG CAG CTA CTA TAC CAC TCG, respectively, and their complements. The fluorescently labeled $parS_F$ substrate was synthesized with Alexa-488 NHS coupled with the 5′ of the forward strand by the manufacturer (IDT).

## Protein purification and fluorescent labeling

For expression of proteins 5 ml of an overnight culture of BL21 DE3 AI (Invitrogen), *E. coli* cells transformed with the desired plasmid were inoculated into 500 ml Terrific Broth (Teknova) supplemented with 100 µg/ml carbenicillin, antifoam Y-40 emulsion (Sigma), 1 g/l NaCl, 0.7 g/l $Na_2SO_4$, 2.6 g/l $NH_4Cl$, and 0.24 g/l $MgSO_4$. The cultures were incubated at 37°C in 2.5 l Fernbach flasks and shaken at 120 rpm until they reached an $OD_{600}$ of 1.8. Cultures were chilled on ice before they were induced by the addition of 1 mM IPTG and 0.2% L-arabinose. Following induction, cultures were incubated at 16°C for 16 hr, and cells were harvested by centrifugation at 6000 × g for 15 min at 4°C. Cell pellets were frozen in liquid nitrogen and stored at −80°C.

Frozen cells were thawed and resuspended to a density of 1 g cell pellet/10 ml in lysis buffer ($ParA_F$, $ParA_F$-eGFP, and $ParB_F^{1-42}$-mCherry-EcoRI$^{E111Q}$: 25 mM Tris–HCl pH 8, 1 M NaCl, 20 mM imidazole, 2 mM β-mercaptoethanol, 10% glycerol; $ParB_F$ and $ParB_F^{R121A}$: 10 mM Sodium Phosphate

buffer pH 7, 1 M NaCl, 20 mM Imidazole, 2 mM β-mercaptoethanol, 10% glycerol; $ParB_F^{1-42}$-mCherry and $ParB_F^{1-42/R36A}$-mCherry: 25 mM HEPES.KOH pH 7.5, 150 mM NaCl, 6 M guanidinium chloride, 20 mM Imidazole, 2 mM β-mercaptoethanol, 10% glycerol) containing EDTA-free Sigmafast protease inhibitor cocktail tablet (Sigma) using a homogenizer. Lysozyme and benzonase (Sigma) were added to a concentration of 1 mg/ml and 50 u/ml, respectively, and the cells were lysed *via* a microfluidizer. Cell debris were pelleted by centrifugation at 142,000 × g for 45 min at 4°C, and the supernatant passed through a 0.22 μm filter.

Lysate was loaded on to a 5 ml HisTrap HP cassette (GE Healthcare) equilibrated in lysis buffer. The cassette was then washed with 10 column volumes of lysis buffer followed by 10 column volumes HisTrap buffer (as lysis buffer without guanidinium hydrochloride and with the following NaCl concentrations: $ParA_F$ proteins, 200 mM; $ParB_F$ proteins, 150 mM), and the protein eluted with a gradient from 20 to 500 mM imidazole over 10 column volumes using an AKTA Pure (GE Healthcare).

All proteins except $ParB_F^{1-42}$-mCherry and $ParB_F^{1-42\ R36A}$-mCherry were then subjected to ion-exchange chromatography. The peak fractions from the HisTrap column were pooled and slowly diluted whilst stirring with a Mono Q/S-buffer (as lysis buffer without imidazole or NaCl, but with 0.1 mM EDTA pH 8) until the conductivity of the sample was as follows: 18 mS/cm for $ParA_F$ proteins, 5 mS/cm for $ParB_F^{1-42}$-mCherry-EcoRI$^{E111Q}$, and 15 mS/cm for all other $ParB_F$ proteins. The conductivity of the samples was monitored using a conductivity meter (Hanna). The sample was loaded onto either a 1 ml Mono Q ($ParA_F$ proteins and $ParB_F^{1-42}$-mCherry-EcoRI$^{E111Q}$) or Mono S (other $ParB_F$ proteins) 5/50 GL ion exchange column (GE Healthcare) pre-equilibrated with Mono Q/S-buffer containing a NaCl concentration to match the conductivity of the sample. The column was then washed with 10 column volumes of Mono Q/S-buffer + NaCl. The protein was eluted with a gradient up to 500 mM NaCl over 10 column volumes.

Finally, all protein samples were purified by size-exclusion chromatography. The peak fractions from the previous column were pooled and diluted 50:50 with concentration buffer (25 mM HEPES. KOH pH 7.5, 2 M NaCl, 2 mM β-mercaptoethanol, 10% glycerol) and concentrated to ~2 ml using a Centriprep 10 kDa spin concentrator (Millipore). The sample was then injected onto an S200 16/600 size exclusion column (GE Healthcare) pre-equilibrated in gel filtration buffer (25 mM HEPES-KOH pH 7.5, 0.1 mM EDTA pH 8, 0.5 mM TCEP and 10% glycerol with 600 mM KCl for $ParA_F$ proteins and 150 mM KCl for $ParB_F$ proteins). Peak fractions were then pooled and concentrated to ~100 μM (~5–10 mg/ml) as determined by UV 280 nm absorbance before being aliquoted, frozen in liquid nitrogen, and stored at –80°C. Protein aliquots were used once and not subjected to freeze-thaw cycles.

To produce fluorescently labeled $ParB_F$ and $ParB_F^{R121A}$, $ParB_F$ protein was buffer exchanged into gel filtration buffer without reducing agent and incubated with a twofold molar excess of Alexa Fluor 647 C2 Maleimide (Thermo Fisher) for 30 min at room temperature. The reaction was then quenched by the addition of DTT to a final concentration of 10 mM. The protein solution was then filtered through a 0.22 μm filter and free dye removed by buffer exchange into gel filtration buffer in Amicon ultra 10 kDa spin concentrator (Millipore). The extent of labeling was estimated based on absorbance at 280 and 647 nm.

## Assaying contaminating activities in the protein preparations

Proteins purified by the above protocol had no significant DNA endonuclease activity. After 16 hr incubation of supercoiled pBR322 with 2 μM $ParA_F$ and/or 10 μM $ParB_F$ at 37°C in ATPase buffer (see below), no linear DNA was observed and less than 10% of the supercoiled plasmid was converted to a nicked-circular form. The contaminating ATPase activity for all $ParB_F$ proteins was less than 2 mol ATP per mol $ParB_F$ per hour, as determined by the ATPase assay protocol detailed below.

## ATPase activity assays

Steady-state ATPase activity was measured as described (*Vecchiarelli et al., 2016*) with modifications. ATP to be used for ATPase activity assays was purified after diluting 20 μCi ATP γ-P$^{32}$ (Perkin-Elmer) in 100 μl of 100 mM unlabeled ATP (Sigma) by passing through a 3 ml P2 resin size-exclusion column equilibrated with a buffer containing 50 mM HEPES·KOH pH 7.5, 150 mM KCl, and 0.1 mM EDTA. The purity of fractions was determined by TLC. One microliter of each fraction was spotted

on to a 10 × 8 cm piece of TLC PEI Cellulose F paper (Millipore) 1 cm above the bottom of the paper and developed for 10 min using 400 mM NaH$_2$PO$_4$ pH 3.6 as the solvent. The fractions containing the minimum contamination of P$^{32}$-Pi were pooled and their concentration determined by spectrometry before storage at −20°C.

ParA$_F$ ATPase activity was measured in the presence of the combinations and concentrations of proteins and DNA cofactors specified in the main text in ATPase buffer (50 mM HEPES·KOH pH 7.5, 150 mM KCl, 5 mM MgCl$_2$, 0.5 mM TCEP, and 1 mM ATP γ-P$^{32}$). Reactions were incubated at 37°C for 4 hr and stopped by the addition of an equal volume of 1 M formic acid. The increase of P$^{32}$-Pi was measured by TLC using PEI Cellulose F paper as detailed above.

## ParB$_F$ NTPase activity assays

Steady-state ParB$_F$ CTPase activity was measured in CTPase buffer containing 50 mM Tris–HCl pH 7.5, 100 mM NaCl, 2 mM MgCl$_2$, 1 mM DTT, 100 µg/ml BSA, 200 µM MESG (EnzChek probe), 1 U/ml of purine nucleotide phosphorylase, and ParB$_F$, parS$_F$ DNA, and CTP at concentrations specified in the figure, following the protocol of the supplier of the EnzChek phosphate assay kit (Thermo-Fisher). Reactions were typically repeated three times using 96-well microtiter plates and the 360 nm absorption signal increase was monitored at 0.5–1 min intervals using Clariostar Plus plate reader (BMG Labtech). The absorption signal increase after subtraction of background time course in the absence of enzyme was converted to released Pi concentration increase based on phosphate titration measurements. The CTP hydrolysis rate was calculated from the initial slope of the time course curve, which typically started after ~7 min deadtime for the plate setting up. Substrate specificity was examined comparing Pi release from four ribonucleoside triphosphates. Attempt to examine inhibition of the CTPase activity by CDP or to detect CDP binding to ParB$_F$ was postponed when the CDP used in this study was found to release Pi upon incubation with ParB$_F$. CDP obtained from two additional suppliers also generated similar quantities of Pi upon incubation with ParB$_F$.

## TIRF microscopy

The general design of the TIRF microscopy setup was essentially as previously described (*Ivanov and Mizuuchi, 2010*; *Vecchiarelli et al., 2013*). A prism-type TIRFM system was built around an Eclipse Ti microscope (Nikon) with a 40× objective (S Fluor, 40×/1.30 oil, Nikon) and two-color images captured by an Andor DU-897E camera through a dxcr630 insert DualView (Photometrics) with the following settings: 3 MHz digitizer (gray scale); 5.2 pre-amplifier gain, 2 MHz vertical shift speed; +one vertical clock range; electron-multiplying gain 30; EM CCD temperature set at −90°C; baseline clamp ON; and exposure time 100 ms.

The excitation for ParA$_F$-eGFP and Alexa647-ParB$_F$ were provided by a 488 nm diode-pumped solid-state laser (Sapphire, Coherent) and a 633 nM HeNe laser (Research Electro-Optics), respectively. The TIRF illumination had an elliptical Gaussian shape in the field of view therefore intensity data for DNA-carpet-bound ParA$_F$-eGFP and Alexa647-ParB$_F$ signals were taken at or near the middle of the illumination profile.

Movies were acquired using Metamorph 7 (Molecular Devices) and transferred to ImageJ (National Institutes of Health) for analysis.

Flow cells were assembled using fused silica microscope slides with pre-drilled inlet/outlet ports (Esco products), #1 glass cover slips (24 × 50 mm, Thermo Fisher) and 0.001'-thick acrylic transfer tape (3M). The fused silica slide was cleaned by soaking overnight in a solution of Nochromix (Sigma)-sulfuric acid, followed by extensive rinsing with de-ionized water, drying by blowing nitrogen gas, followed by oxygen plasma treatment (South Bay Technology Inc). The Y-shaped flow path pattern was cut out of the transfer tape using a laser cutter before the flow cell assembly. Nanoports (Idex) were attached to the fused silica slide for the inlet and outlet tube connections using Norland Optical Adhesive (Thorlabs), cured by 365 nm UV light. The assembled flow cells were then baked at 80°C with gentle compression for 2 hr.

To assemble a DNA-carpet in a flow cell, small unilamellar vesicles (SUVs) of 1,2-dioleoyl-*sn*-glycero-3-phosphocholine (DOPC) and 1,2-dioleoyl-*sn*-glycero-3-phosphoethanolamine-N-(biotinyl) (DOPE-Biotin) (Avanti Polar Lipids) were prepared as follows. 0.5 ml of DOPC (25 mg/ml chloroform) was mixed with 5 µl of DOPE-biotin (25 mg/ml chloroform) in a glass test tube and most of the solvent removed *via* evaporation under a nitrogen flow. The remaining solvent was removed by drying

in a SpeedVac (Savant) at 42°C for 1 hr followed by a further 1 hr at room temperature. 2.5 ml of degassed TK150 buffer (25 mM Tris–HCl pH 7.5, 150 mM KCl) was then added, and the lipids stored and covered under nitrogen gas overnight. The lipids were then resuspended by vortexing and sonicated (70–80 watts, 30 s on, 10 s off) in a cup horn with water chiller set to 16°C (QSonica) until transparent. The resulting solution of SUVs was then filtered through a 0.22 µm filter, aliquoted, and stored under nitrogen gas at 4°C for up to 4 weeks.

To prepare biotinylated salmon sperm DNA for DNA-carpets 10 mg/ml salmon sperm DNA (Thermo Fisher) was sonicated for 5 min (110 watts, 10 s on, 10 s off) to produce short fragments. Sonicated salmon sperm DNA was then diluted to 1 mg/ml in Terminal Transferase buffer (NEB) with 0.25 mM $CoCl_2$, 40 µM Biotin-14-dCTP (Thermo Fisher), and 1 unit/µl Terminal Transferase (NEB). The DNA was incubated at 37°C for 30 min, and then the reaction stopped by heat inactivation at 75°C for 20 min. Free Biotin-14-dCTP was removed by extensive buffer exchange with TE buffer (10 mM Tris–HCl pH 8, 0.1 mM EDTA) in a 100 kDa Amicon Ultra spin concentrator (Millipore). The biotinylated DNA was then concentrated to ~10 mg/ml and stored at −20°C until needed.

To assemble a DNA-carpet, the DOPC–DOPE-biotin SUV solution was diluted to 1 mg/ml in 500 µl degassed TN150MC buffer (25 mM Tris–HCl pH7.5, 150 mM NaCl, 5 mM $MgCl_2$, 0.1 mM $CaCl_2$) and warmed to 37°C. Approximately 300 µl of SUV solution was then infused into a pre-warmed flow cell and incubated at 37°C for 1 hr. Excess SUVs were washed out with 500 µl warmed, degassed TN150MC buffer at 100 µl/min. 300 µl of a solution of 1 mg/ml neutravidin (Thermo Fisher) in warmed, degassed TN150MC buffer was then infused at a rate of 100 µl/min into the flow cell and incubated at 37°C for 30 min. Excess neutravidin was washed out with TN150MC buffer as above, and the flow cell infused with 100 µl of a solution containing 1 mg/ml biotinylated sonicated salmon sperm DNA (as prepared above) in warmed, degassed TN150MC buffer, and incubated at 37°C for 30 min. The ports of the flow cell were sealed with parafilm and stored at 4°C for up to a week.

Prior to use, excess DNA was removed by infusion of 300 µl 0.22 µm filtered and degassed TIRFM buffer (50 mM HEPES–KOH pH 7.5, 300 mM K-glutamate, 50 mM NaCl, 10 mM $MgCl_2$, 0.1 mM $CaCl_2$, 2 mM DTT, 0.1 mg/ml α-casein, 0.6 mg/ml ascorbic acid, 10% glycerol) with addition of 1 mg/ml α-casein and 1 mM ATPγS and the flow cell incubated at room temperature for 30 min.

Conversion of the fluorescence signal detected in TIRF microscopy to the DNA-carpet-bound protein densities was done following the procedure described in the legend of Figure 2—figure supplement 4 in *Vecchiarelli et al., 2016*.

## Fluorescence recovery after photobleaching (FRAP)

For FRAP experiments, 488 nm solid-state and 630 nm diode lasers were focused to the back focal plane of the objective through an appropriate dichroic mirror (Di01-R405/488/561/635-25x36, Semrock) through the objective lens to illuminate a ~5 or ~10 µm (for 488 nm or 630 nm, respectively) diameter spot in the center of the sample area. The laser power was adjusted for ~80% bleaching with 5 s exposure for the eGFP or Alexa 647 signals, and four cycles of bleaching/recovery were recorded for each sample and averaged.

## Magnetic tweezers-based DNA condensation assay

The magnetic tweezers setup and assays conducted with it were performed as previously described (*Seol and Neuman, 2011*; *Seol et al., 2016*).

The ability of ParB$_F$ to condense *parS$_F$*-containing DNA (spDNA) was tested by a custom-built magnetic tweezers setup. In brief, two permanent magnets were used to apply force to micron-sized magnetic beads individually tethered to the coverslip of a one inlet flow cell by 5 kb pET28a plasmid-derived DNA tethers. The distance the magnets were held from the beads, and hence the force exerted upon them was controlled by a linear motor that vertically positions the magnets.

Five kilo-base DNA substrates were generated by PCR using either pET28a-*parS$_F$* plasmid (for *parS$_F$*-containing DNA) or pET28a as templates. pET28-*parS$_F$* DNA plasmid was generated by cloning 570 bp DNA segment containing 12 repeats of *parS$_F$* native sequence from F-plasmid into pET28a between the BamHI and SphI restriction sites.

Primers used for the PCR contained an extra non-complementary 15 nt at their 5′ ends to encode BsaI restriction sites. The PCR yields a 5.2 kb product incorporating two BsaI restriction sites at its termini. Digestion of this product was followed by ligation with 500 bp DNA 'handles' containing

either multiple biotin or digitoxin labels. These handles were also generated by Taq-based PCR using pBlueScript II KS as the template, pBlueScript II KS forward (5'- GCT GGG TCT CGG TTG TTC CCT TTA GTG AGG GTT AAT TG) and pBlueScript II KS reverse (5'- GCT GGG TCT CGT GGT TTC CCT TTA GTG AGG GTT AAT TG) primers and either 60 µM biotin-16-dUTP or digoxigenin-11-dUTP (Roche). This results in a 5 kb DNA tether, which can be attached to a streptavidin-coated magnetic bead at one end and an anti-digoxigenin coverslip surface at the other.

ParB$_F$ samples were prepared in modified ATPase buffer (50 mM HEPES.KOH pH 7.5, 100 mM KCl, 5 mM MgCl$_2$, 2 mM DTT, 10 mg/ml BSA, and 0.1% Tween-20) and infused into a flow cell containing tethered magnetic beads held at 5 pN of force. After the chamber was filled, the flow was stopped, and the force reduced to 0.05 pN. The height of beads was tracked by analysis of diffraction rings generated by illumination of the beads from above and observed with an objective positioned below the flow cell. The extent of condensation by ParB was monitored by the decrease in the height of the beads at 0.05 and 5 pN as compared to controls without protein.

## Acknowledgements

We are grateful to helpful suggestions and discussion of our colleagues Barbara Funnell, David Lane, Michiyo Mizuuchi, Andrea Volante, Min Li, Masaki Osawa, William Carlquist, and Shannon Mckie, to Esme Neuman for help in preparation of *Figures 1* and *7*, and to Min Li for help in preparation of *Figure 2—figure supplement 3*. We thank Stephan Gruber and his colleagues for sharing their findings with us prior to publication. This work was supported by the intramural research fund for National Institute of Diabetes and Digestive and Kidney Diseases (KM), and the National Heart, Lung, and Blood Institute (KCN), National Institutes of Health, Department of Human Services.

## Additional information

### Funding

| Funder | Grant reference number | Author |
| --- | --- | --- |
| NIDDK | Intramural Research Fund | Kiyoshi Mizuuchi |
| NHLBI | Intramural Research Fund | Keir C Neuman |

The funders had no role in study design, data collection and interpretation, or the decision to submit the work for publication.

### Author contributions

James A Taylor, Conceptualization, Data curation, Formal analysis, Writing - original draft, Writing - review and editing; Yeonee Seol, Data curation, Formal analysis, Writing - original draft, Writing - review and editing; Jagat Budhathoki, Data curation, Formal analysis; Keir C Neuman, Kiyoshi Mizuuchi, Conceptualization, Resources, Formal analysis, Supervision, Funding acquisition, Writing - review and editing

### Author ORCIDs

Keir C Neuman (ID) http://orcid.org/0000-0002-0863-5671
Kiyoshi Mizuuchi (ID) https://orcid.org/0000-0001-8193-9244

### Decision letter and Author response

Decision letter https://doi.org/10.7554/eLife.65651.sa1
Author response https://doi.org/10.7554/eLife.65651.sa2

## Additional files

### Supplementary files

• Source data 1. Excel file containing all source data. One Excel file with 19 sheets: Numerical data for *Figure 2A-E*, *Figure 2F,G*, *Figure 2—figure supplement 2*, *Figure 2—figure supplement 4*,

*Figure 3A-C, Figure 3D, Figure 4A-D, Figure 4E, Figure 4—figure supplement 1B, Figure 4—figure supplement 2A-D, Figure 5A-D, Figure 5E,F, Figure 5—figure supplement 1B, Figure 5—figure supplement 2, Figure 6A, Figure 6B - left, Figure 6B - right, Figure 6C, Figure 6—figure supplement 1A, Figure 6—figure supplement 1B, Figure 6—figure supplement 2, Figure 6—figure supplement 3.*

- Transparent reporting form

## Data availability

All data generated or analysed during this study are included in the manuscript and supporting files. Source data files have been provided for all relevant figures.

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
