## [Decision Letter]

**Acceptance summary:**

This manuscript reports carefully executed experiments on the dynamics of ParA-ParB and ParB-ParB interactions. Two main findings are presented: a change in stoichiometry of ParA-ParB interactions upon ligand binding and ligand dependent DNA condensation by ParB. The authors have thoroughly and appropriately responded to all comments. The textual changes and additional experiments (e.g. with relaxed DNA) have significantly improved the manuscript. The paper paves the way for future research on the intricate ParA-ATP ParB-CTP parS interplay.

**Decision letter after peer review:**

Thank you for submitting your article "CTP and *parS* control ParB partition complex dynamics and ParA-ATPase activation for ParABS-mediated DNA partitioning" for consideration by *eLife*. Your article has been reviewed by 2 peer reviewers, and the evaluation has been overseen by a Reviewing Editor and Volker Dötsch as the Senior Editor. The following individual involved in review of your submission has agreed to reveal their identity: Stephan Gruber (Reviewer #1).

Essential revisions:

1) The interaction between ParBF and CTP itself (in the presence/absence of parSF) has not received much attention from the authors. How strong does ParBF bind to CTP? Does ParBF bind CTP in the absence of parSF? How fast does ParBF hydrolyze CTP in the presence/absence of parSF? This information is very important to fully assess the results in this manuscript, especially those where all components: ParAF, ParBF, ATPyS, CTP, parSF are present. (Some of the initial information is already in Soh et al. 2019, but those were preliminary observations.)

2) According to the presented model, the stoichiometry of ParA-ParB interactions in the presence of CTP and/or parS could be changed by providing additional flexibility in the connection between the N-terminal ParA-interacting peptide in ParB and the ParB CTP domain. If indeed the stoichiometry is critical, then such ParB mutants are expected to feature plasmid partitioning defects. If this exp can be performed, it would be important to show as it will be directly relevant to the most important finding in this manuscript i.e. the stoichiometry of ParA-B interaction +/- CTP.

Other major concerns worth addressing if feasible:

3) The work reported in this manuscript relies on the recruitment of ParA-ATPgS to DNA, which is apparently inefficient when compared to ParA-ATP (Vecchiarelli, 2013?). This fact (if indeed true) needs to be stated more explicitly in the text, currently it is only indirectly implied. Is there a way to test selected findings in a more physiological setting using ATP (e.g. with ParA active site mutants)?

4) The results from the ParB DNA condensation experiments are intriguing. Could the authors estimate/determine the (local) concentration of ParB needed for DNA compaction? Similarly, it would be helpful to investigate whether progressive condensation of DNA occurs indeed by sequential recruitment of ParB dimers to the parS site (Figure 6A). Would DNA compaction arrest upon blockage of parS?

5) From the magnetic-tweezers (MT) experiments, Taylor et al. suggested that "Our results show that DNA-condensation is caused by ParBF-ParBF interactions forming DNA-looping bridges without requiring other protein factors". The dominant view of the field, before the discovery of CTP, is that ParB-ParB forms protein-mediated bridges that condense DNA. Now that we know ParB most likely form a sliding clamp on DNA, there is a possibility that the sliding causes DNA supercoiling, akin to how RNAP/PNCA runs on DNA, and that DNA supercoiling causes the reduction in DNA extension in the MT experiment. Can Taylor et al. distinguish between the two possibilities? May be repeating the MT experiments with a nicked DNA to dissipate any accumulated supercoiling?

6) Does ParBF bind CDP in the presence/absence of parSF? Experiments in this manuscript suggest that it does (Figure 5F, Figure 6B-C), but other chromosomal ParBs are not known to bind to CDP.

7) The other intriguing area is how ParBF CTPase activity is affected by the presence of ParA and this in itself has important implications for the "diffusion-ratchet" model. Taylor et al. might want to investigate this in the future, but maybe the authors can mention this in the Discussion?

---

## [Author Response]

Essential revisions:1) The interaction between ParBF and CTP itself (in the presence/absence of parSF) has not received much attention from the authors. How strong does ParBF bind to CTP? Does ParBF bind CTP in the absence of parSF? How fast does ParBF hydrolyze CTP in the presence/absence of parSF? This information is very important to fully assess the results in this manuscript, especially those where all components: ParAF, ParBF, ATPyS, CTP, parSF are present. (Some of the initial information is already in Soh et al. 2019, but those were preliminary observations.)

Thank you for pointing out this issue. We had preliminary data for the ParB_F_ CTPase activity, but we did not have publishable data. We carried out a new set of CTPase measurements in the presence and absence of *parS_F_* to estimate the apparent CTP *K_M_* and *k_cat_*, which are now presented in Figure 5—figure supplement 2, and described on page 15, lines 295-304 (line numbers are for Print Layout View without Markup display).

We are also very much interested in the characterization of the ParB_F_ CTPase activity in the presence of ParA_F_ and other cofactors. However, we wish to approach this subject in a systematic manner, if possible, including the study of the ParB CTPase activity in the presence of ATP hydrolysis by ParA. However, because g-^32^P-CTP is not readily available (and removal of contaminating CDP from a-^32^P-CTP is not easy compared to removal of π from g-^32^P-ATP as done for our ATPase assay substrate), we are currently evaluating the technical options available to us. Completing this project under current working conditions will take a substantial length of time and incorporating this study would not only cause lengthy delay in publication of the current manuscript, but also make it excessively long. Therefore, we plan to carry out a comprehensive study separately to be published in a follow up paper.

2) According to the presented model, the stoichiometry of ParA-ParB interactions in the presence of CTP and/or parS could be changed by providing additional flexibility in the connection between the N-terminal ParA-interacting peptide in ParB and the ParB CTP domain. If indeed the stoichiometry is critical, then such ParB mutants are expected to feature plasmid partitioning defects. If this exp can be performed, it would be important to show as it will be directly relevant to the most important finding in this manuscript i.e. the stoichiometry of ParA-B interaction +/- CTP.

Thank you for pointing out this important test of the model presented here. We indeed have been considering construction of such artificial mutants to support the hypothesis.

However, the N-terminal region of plasmid ParBs such as F or P1 are longer than chromosomal ParBs, and the boundary between the ParA-interaction domain and the CTPase domain remains somewhat ambiguous, necessitating a few preliminary experiments before we can finalize the design of the mutant protein. Since we are still limited in our capacity to carry out experiments, inclusion of these experiments in the current manuscript would cause substantial delay in publication. In the revised manuscript we included predictions for such mutants and provide an overview of our future plans in relation to these mutants in the Discussion (page 27, lines 535-540).

Other major concerns worth addressing if feasible:3) The work reported in this manuscript relies on the recruitment of ParA-ATPgS to DNA, which is apparently inefficient when compared to ParA-ATP (Vecchiarelli, 2013?). This fact (if indeed true) needs to be stated more explicitly in the text, currently it is only indirectly implied. Is there a way to test selected findings in a more physiological setting using ATP (e.g. with ParA active site mutants)?

ParA-ATPgS is indeed compromised in the conversion from non-DNA binding dimer to DNA-binding dimer (Vecchiarelli 2010, 2013), and we utilized this property to our advantage in this study to ensure that the nsDNA-bound proteins represent those forming the complex. We added clarifications of this point in the revised manuscript (Page 7, lines 133-136).

However, the difference in the nsDNA binding property of ParA-ATPgS and ParA-ATP appears subtle. For example, at higher ParA concentrations ParA-ATPgS binds DNA more efficiently without ParB. It is possible some ATPase active site mutant could be found that would allow us to carry out experiments in the presence of ATP. However, to date we have not found a mutant that renders DNA binding dependent on ParB in the presence of ATP. Use of the mutant ParA that binds nsDNA in the presence of ATP without ParB would require a new experimental protocol, which is yet to be established. Therefore, we will wait for the next stage of the project to pursue possible use of the ATPase-defective ParA in the characterization of the ParA-ParB complex.

4) The results from the ParB DNA condensation experiments are intriguing. Could the authors estimate/determine the (local) concentration of ParB needed for DNA compaction? Similarly, it would be helpful to investigate whether progressive condensation of DNA occurs indeed by sequential recruitment of ParB dimers to the parS site (Figure 6A). Would DNA compaction arrest upon blockage of parS?

Thank you for pointing out this important limitation in our current experiment. In this study, the magnetic tweezers instrument was not equipped with fluorescence imaging capability so we were not able to quantitate ParB_F_ spreading around the *parS_F_* site during DNA condensation. Therefore, we cannot directly estimate the number of ParB_F_ molecules involved in condensing DNA. However, in line with recently published findings from other groups concerning chromosomal ParBs, we believe our results are consistent with the notion that the tightly condensed state of the *parS_F_*-containing DNA in our experiments contains a large number of ParB_F_ molecules. Our results as well as published work from other groups predict that blockage of parS would arrest DNA compaction. However, this has not been directly tested. In the revised manuscript, we added a statement to clarify these points in the Discussion on page 23, lines 448-452. We are currently preparing an experimental design for the next stage of our study to address a number of unanswered questions including the points raised by the reviewers here.

5) From the magnetic-tweezers (MT) experiments, Taylor et al. suggested that "Our results show that DNA-condensation is caused by ParBF-ParBF interactions forming DNA-looping bridges without requiring other protein factors". The dominant view of the field, before the discovery of CTP, is that ParB-ParB forms protein-mediated bridges that condense DNA. Now that we know ParB most likely form a sliding clamp on DNA, there is a possibility that the sliding causes DNA supercoiling, akin to how RNAP/PNCA runs on DNA, and that DNA supercoiling causes the reduction in DNA extension in the MT experiment. Can Taylor et al. distinguish between the two possibilities? May be repeating the MT experiments with a nicked DNA to dissipate any accumulated supercoiling?

The reviewer suggests an interesting point concerning the possibility of compaction being driven in part by supercoiling of the DNA. To address this insightful possibility, we have included a comparison of the compaction of nicked DNA versus supercoilable DNA by ParB. We could distinguish nicked DNA from “coilable” (rotationally constrained) DNA by twisting the individual DNA molecules in the magnetic tweezers prior to ParB_F_ binding. Twisting intact coilable DNA results in plectoneme formation, evidenced by a decrease in the DNA extension, whereas the extension of nicked or torsionally unconstrained DNA is insensitive to twisting. We found that condensation by ParB was comparable for both rotationally constrained and unconstrained DNA molecules, suggesting that DNA condensation by ParB is unlikely due to “effective supercoil generation by ParB moving along DNA”. In the revised manuscript, we added this result as Figure 6—figure supplement 2 and added text explaining this on page 19, lines 355-358.

6) Does ParBF bind CDP in the presence/absence of parSF? Experiments in this manuscript suggest that it does (Figure 5F, Figure 6B-C), but other chromosomal ParBs are not known to bind to CDP.

Our earlier results suggested that CDP (2 mM) likely binds ParB_F_ under our reaction conditions. However, the difference from the results of experiments without C-nucleotides was relatively subtle. As a response to the reviewer’s comment, we attempted to examine CDP binding to ParB_F_ by biophysical methods and also tried to measure the apparent *K_i_* of CDP for the CTPase activity. However, we encountered technical difficulties, which we eventually determined arose from the fact that the CDP preparation we used was contaminated by a small but significant quantity (~2-3%) of a compound that acts as a substrate of ParB_F_, releasing Pi (very likely to be CTP, but we have not yet directly determined the identity of the contamination). We examined several batches of CDP obtained from several sources, but failed to identify a CDP preparation free of this contamination. We have not been able to locate a commercial source of higher quality CDP and purification of CDP under current lab operating conditions would result in further substantial delays. In addition, even if we succeed in obtaining CDP essentially free of this contaminant, some of the experiments in the paper that involved CDP are currently not readily repeatable in a timely fashion; the first author who carried out these experiments had to leave the country last year and will not be able to return to repeat the experiments, and the instrument used for the study is currently undergoing reconfiguration to address a failed laser illumination system.

Fortunately, in the current paper the experiments that involved CDP were done essentially as control experiments to be compared with the results of experiments involving CTP, and the impacts of the relatively low concentration of the contaminant, if any, appears relatively minor. In other words, the results of the experiments involving CDP were clearly different from those in the presence of CTP but not dramatically different from those without C-nucleotides, indicating that either CDP does not significantly bind ParB_F_ under the conditions used, or if bound, it likely does not significantly change the behavior of ParB_F_ for the properties examined in most of the experiments. Therefore, the above finding did not appear to have substantial impacts on the general conclusions of the current paper. We described this finding on page 16, lines 305-312 (and Figure 5—figure supplement 2D). In order to avoid confusions we also eliminated a small fraction of the data (Figure 5B, Figure 5F- CDP data in the original manuscript), interpretation of which are ambiguous in light of this finding. We edited the manuscript at number of additional places to reflect this new finding, such as page 19, lines 362-364.

We note that Osorio-Valeriano et al., showed ParB_MX_ binds CDP with a low affinity, but their measurement also used CDP supplied by Sigma with the same catalog number as used in our experiments, which was found to contain the contaminant. Therefore, the published *K_D_* for CDP may not be accurate.

We thank the reviewers for the comment, which prompted us to examine the CDP-ParB interaction and led us to find the problem we were unaware of. We are planning on purification of the nucleotides when the required equipment becomes accessible to us.

7) The other intriguing area is how ParBF CTPase activity is affected by the presence of ParA and this in itself has important implications for the "diffusion-ratchet" model. Taylor et al. might want to investigate this in the future, but maybe the authors can mention this in the Discussion?

As mentioned in the response to major comment 1), we are very interested in expanding our study to address the questions suggested here. Ideally, a sensitive and nucleotide-specific CTPase assay in the presence of ATPase activity in the reaction would be nice. If this turns out to be difficult, we may start the study under conditions where ATP hydrolysis does not take place either by using ATPgS or a mutant ParA.

For the current manuscript, as suggested, we mentioned this point in the Discussion as one of several remaining questions to be addressed by future study on page 27, lines 544-545.